



# Regionalizing non-parametric precipitation amount models on different temporal scales

Tobias Mosthaf[1] and András Bárdossy[1]

[1]Institute for Modeling Hydraulic and Environmental Systems, Universität Stuttgart, Stuttgart, Germany

*Correspondence to:* Tobias Mosthaf (tobias.mosthaf@iws.uni-stuttgart.de)

**Abstract.** Parametric distribution functions are commonly used to model precipitation amounts. Precipitation amounts themselves are crucial for stochastic rainfall generators or weather generators. Non-parametric kernel density estimates (KDEs) offer a more flexible way to model precipitation amounts. As it is already stated in their name, these models do not exhibit a parameter which can easily be regionalized to run rainfall generators at ungauged locations as well as at gauged locations. To overcome this deficiency we present a new interpolation scheme for non-parametric models and evaluate it for different temporal resolutions ranging from hourly to monthly. During the evaluation the non-parametric methods are compared to commonly used parametric models like the two parameter gamma and the mixed exponential distribution. As water volume is considered to be an essential parameter for applications like flood modeling, a Lorenz-curve based criterion is also introduced. To add value to the estimation of data at sub-daily resolutions, we incorporated the plentiful daily measurements in the interpolation scheme and this idea was evaluated. The study region is the federal state of Baden-Württemberg in the southwest of Germany with more than 500 rain gauges. The validation results show that the newly proposed non-parametric interpolation scheme works, and additionally seems to be more robust compared to parametric interpolation schemes, and that the incorporation of daily values in the regionalization of sub-daily models is very beneficial.

## 1 Introduction

Rainfall time series of differing temporal resolutions are needed for various applications like water engineering design, flood modeling, risk assessments or ecosystem and hydrological impact studies (Wilks and Wilby, 1999; Burton et al., 2008). As many precipitation records are too short and contain erroneous measurements, stochastic precipitation models can be used to generate synthetic time series instead. Starting from single site models (summarized in Wilks and Wilby, 1999), multi site models for simultaneous time series at various sites (e.g., Wilks, 1998; Buishand and Brandsma, 2001; Bárdossy and Plate, 1992) and finally models which allow for gridded simulations are developed (e.g., Wilks, 2009; Burton et al., 2008).

For modeling precipitation one crucial variable is the precipitation amount, which follows a certain distribution. Distributions of daily precipitation amounts are strongly right skewed, with many small values and few large values (Wilks and Wilby, 1999; Li et al., 2012; Chen and Brissette, 2014). This also holds true for different temporal resolutions with increasing skewness for higher temporal resolutions and vice versa. This means, that rainfall intensity distributions depend on the temporal scale of the observed values. In order to run single site precipitation models at ungauged locations, or for their gridded applications,





precipitation amount distributions are also needed for these sites. To reach this goal, regionalization is required which can be done in two different ways:

1. Interpolate precipitation amounts from observation points for every time step to the target location(s) and set up a distribution with the interpolated values.

2. Fit a distribution function to the precipitation amounts separately for each gauge and interpolate the distribution functions to the target location(s).

The first approach is maybe more straightforward, but exhibits several deficiencies such as overestimation of the rainfall probability, underestimation of the variance and underestimation of the maximum rainfall value. In section 9, an example will demonstrate these problems. Due to the relative inefficiency of the first interpolation approach the second is preferred.

10 In most stochastic rainfall models, theoretical parametric distribution functions are fitted to the empirical values using e.g. the exponential distribution or the two parametric gamma distribution (Wilks and Wilby, 1999; Papalexiou and Koutsoyiannis, 2012). It is possible to either interpolate the parameters of the theoretical distribution or to interpolate the moments (e.g. mean and standard deviation) of the rainfall intensities (Wilks, 2008; Haberlandt, 1998). Lall et al. (1996) introduced a more flexible non-parametric rainfall model, where they used non-parametric KDEs with a prior logarithmic transformation to model daily

15 rainfall intensities. They address the problem of regionalization by using non-parametric estimates of distribution functions, as they do not use any parameter, which can simply be interpolated.

In the present work we introduce a regionalization strategy for non-parametric distributions and compare it to the traditional regionalization of parametric distributions for varying temporal resolutions from hourly to monthly scale. The common procedure to interpolate parametric distribution functions is:

1. Fit a parametric distribution (e.g. a gamma or exponential distribution) at each sampling site to the empirical distribution function (EDF).

2. Interpolate the moment(s) or parameter(s) of the fitted parametric distribution.

3. Set up the theoretical cumulative distribution function (CDF) at every interpolation target with the interpolated moment(s) or parameter(s).

25 The newly proposed procedure for non-parametric distribution functions is:

1. Fit a non-parametric distribution to log-transformed rainfall values using a Gaussian kernel.

2. Estimate the interpolation (kriging) weights with the precipitation values of a certain quantile.

3. Apply these weights to the values of certain discrete quantiles.

4. Linearly interpolate the remaining quantile values to receive a continuous CDF for all target locations.





In Arns et al. (2013) a similar approach is used to interpolate quantile value differences of water levels for a bias correction between empirical distributions of observed and modeled values at the German North Sea Coast. In contrast to their work, entire theoretical distribution functions through interpolation are estimated in our work. Goulard and Voltz (1993) introduced a curve kriging procedure to regionalize fitted functions, which was further developed by Giraldo et al. (2011). As CDF curves

are special functions, which are monotonically non-decreasing between 0 and 1, the curve kriging procedure would have been additionally constrained to these conditions. Our approach can deal with these conditions directly.

After describing the study region Baden-Württemberg in section 2, the concept of precipitation amount models will be introduced in section 3. The data selection in section 4 is followed by an investigation of the spatial dependence of the precipitation amount models in section 5. The theory of precipitation amount models will be addressed in section 6 and the basis of

the proposed interpolation procedure for non-parametric models will be established in section 7. The application of different regionalization procedures of precipitation amount models will be explained in section 8 and motivated in the regionalization example in section 9. How to make use of daily precipitation values will be demonstrated in section 10. The resulting performance of different precipitation amount models at point locations and their regionalization is depicted in section 11.

## 2   Study region and data

The study region is the federal state of Baden-Württemberg, which is located in the southwest of Germany. The mountain range Black Forest in the western part and the mountain range Swabian Alps extending from southwest to northeast exhibit the highest elevations in Baden-Württemberg. The rising of large scale moist air masses across mountainous regions causes higher rainfall amounts on the windward side and lower amounts on the leeward side. In the summer months slopes with differing inclinations lead to a warming of the air triggering convection currents, which leads to a greater number of showers

and thunderstorms over mountainous regions. This shows a dependence of rainfall on elevation with seasonal differences. The rain-bearing westerly winds lead to high rainfall amounts in the Black Forest. The less high mountains of the Swabian Alps only exhibit relatively low rainfall amounts as they lie in the shadow of the Black Forest (Landesanstalt für Umwelt, Messungen und Naturschutz Baden-Württemberg (LUBW), 2006).

As an investigation period the years from 1997 - 2011 are chosen, as the German Meteorological Service (DWD) set up

many new rain gauges in 1997. To obtain a relatively homogeneous dataset, only gauges with at least five years of data and a data availability of at least 80 % are chosen. It turned out that we had access to (i) 242 hourly and 5 min resolution and (ii) 347 daily gauges available in the study region, with 80 sites having both high and daily resolution instruments. The observations are provided by the DWD and the Environmental Agency of Baden-Württemberg (LUBW). The high resolution rain gauges are mostly equipped with tipping buckets and gravimetric measurement devices (Beck, 2013). Fig. 1 shows the study region

with the locations of the two sets of rain gauges.



## 3 Modeling precipitation amounts at point locations

Modeling precipitation amounts in our context means estimating distribution functions. Using distribution functions includes the implicit assumption of temporally independent and identically distributed (i.i.d.) variables. This assumption is generally accepted for daily rainfall as the autocorrelation of consecutive nonzero daily precipitation is relatively small and usually of

less importance. For higher temporal resolutions like the hourly one, autocorrelation needs to be incorporated in the model (Wilks and Wilby, 1999). In practice different methods exist to take account of this correlation. One approach is to include autocorrelation prior to the sampling procedure by using conditional distributions. Conditions may be event statistics like the duration of a rainfall event (e.g., Acreman, 1990) or varying statistical moments depending on the hour of the day (e.g., Katz and Parlange, 1995). Another approach is introducing autocorrelation after the sampling procedure. Bárdossy (1998) uses

empirical distributions of hourly rainfall intensities to sample values whose random order is subsequently changed within a Simulated Annealing scheme to consider autocorrelation. In Bárdossy et al. (2000) theoretical representations of the empirical distributions are used to allow regionalization of the distributions and enable simulations at ungauged locations.

In general, a point wise distribution of precipitation amounts is fully described by its CDF. The CDF $F$ of a random variable $X$ is (DeGroot and Schervish, 2012)

$$F(x) = P(X \leq x) \qquad for \quad -\infty < x < \infty, \tag{1}$$

where $P$ stands for (non-exceedance) probability and the random variable $X$ represents precipitation amount values at a given location. For observed rainfall values the EDF can be obtained directly from observed values. To obtain a continuous CDF for observed rainfall values a theoretical distribution is fitted to the EDF. In addition to enabling regionalization, further advantages of smooth estimates of empirical distributions are described in Lall et al. (1996). Sampling from smooth estimates prevents

the repetition of observed values and offers the opportunity to sample continuous values instead of discrete values. The values $F(x)$ of the CDF are referred to as *quantiles* in this work and the $x$ values are called *quantile values*.

## 4 Data selection

For applications of rainfall estimates like hydrological or hydraulic modeling the correct representation of small rainfall values is not necessary as their contribution to decisive high discharge rates is rather small. Furthermore, tipping bucket gauges lead to

wrong estimates especially for low rainfall values (Habib et al., 2001). Relative estimation errors are increasing for decreasing rainfall rates (Nystuen et al., 1996; Ciach, 2003) and they only represent a small part of the total water volume, but the number of smaller rainfall values is rather high. To avoid the negative effect of this high number of inaccurate values and due to their minor importance for further applications, this study focuses on medium and high rainfall values.

Therefore, the quantile threshold ($Q_{th}$) for the one hourly (1H) values is set to 0.95. This means, that values smaller than the

0.95 quantile value are excluded. To investigate the total water volume represented by rainfall values above the 0.95 quantile at point locations, the Lorenz-curve (Lorenz, 1905) is used. We considered a water volume analysis for varying quantiles as important, to show that high quantiles not only represent the decisive higher rainfall values, but also a large proportion of the





total water volume. So focusing on these quantiles during the model setup is likely to lead to a better model as lower quantiles would disturb the model estimation due to measurement errors and the higher quantiles already represent a great percentage of the total water volume. The volume of the lower quantiles can then be modeled by simple and robust methods as they do not require a very precise estimation due to their high inaccuracy and minor importance.

The Lorenz-curve was introduced by Lorenz (1905) to measure the concentration of wealth. After arranging the $n$ observations $x_i$ in non-decreasing order, the Lorenz-curve $L_i$ can be calculated from a population (in our case rainfall values at a single gauge) with the following formula:

$$L(i) = \frac{\sum_{j=1}^{i} x_j}{\sum_{j=1}^{n} x_j} \qquad (2)$$

The hourly threshold quantile values ($QV_{th}$) range between 0.2 and 1.7 mm for the 0.95 $Q_{th}$ depending on the location of the gauge (see Table 1). The Lorenz-curve of hourly values above $Q_{th}$ in Fig. 2 a shows that their relative water volume is between 70 and 95 %.

Based on hourly values (1H) of the high resolution data set, aggregated rainfall values of different temporal resolutions are obtained: 2 hourly (2H), 3 hourly (3H), 6 hourly (6H) and 12 hourly (12H). Through aggregation of daily values (1D) of the daily data set, 5-daily (5D) and monthly (M) values are obtained. In order to exclude small values and still consider the values producing a high percentage of the water volume, the $Q_{th}$ for sub-daily resolutions are defined with the mean Lorenz-curves in Fig. 2 b. 85% is defined as target percentage of the water volume with the value of the mean hourly Lorenz-curve for the 0.95 $Q_{th}$. This target percentage yields the following $Q_{th}$ for sub-daily resolutions: 0.93, 0.92, 0.9, 0.86 (see Table 1). For aggregations greater or equal to one day, the number of values is rather small and there estimation errors are lower due to an increasing accumulation time (Ciach, 2003; McMillan et al., 2012). Nevertheless, only values above the highest quantile of 1 mm in the study region are used for the daily (1D) and 5-daily (5D) resolution (see Table 1), as smaller values may still exhibit measurement errors.

For the estimation of these basic statistics in Table 1 and for following calculations, rain values of the investigated aggregations smaller than 0.1 mm are set to 0 mm. The reason is to achieve homogenization of the data sets of different years and gauges, as the discretization ranges from 0.01 mm to 0.1 mm depending on the gauge.

## 5 Probability distributions of precipitation amounts in a spatial context

This section focuses on the spatial dependence of precipitation amount distributions, as the applied interpolation technique of ordinary kriging (OK), is based on the assumption that the variable of interest (the CDF) exhibits decreasing similarity with increasing distances. For the purpose of describing the development of the distribution functions in space, the test criterion $T$ of the two-sample Cramér–von Mises criterion is used (Anderson, 1962). It evaluates the similarity of two CDFs, in our case the similarity of CDFs from observations of two different point locations. The test criterion $T$ is defined according to Anderson (1962) as:

$$T = \frac{U}{NM(N+M)} - \frac{4MN-1}{6(M+N)} \qquad (3)$$





where

$$U = N \cdot \sum_{i=1}^{N} (r_i - i)^2 + M \cdot \sum_{j=1}^{M} (s_j - j)^2 \qquad (4)$$

with $N$ as number of observations of the first sample, $M$ as number of observations of the second sample. Both observations are joined together in one pooled dataset and the ranks are determined in ascending order of all observations in the pooled dataset. $r_i$ are the ranks of the $N$ observations of the first sample in the pooled dataset and $s_j$ are the sorted ranks of the $M$ observations of the second sample in the pooled dataset. $T$ can be interpreted as the mean difference of CDF values (quantiles) of observed rainfall values between both data sets. So, if $T$ increases for increasing distances, the CDFs are less similar for increasing distances.

For the calculations of $T$, only rainfall values above the different $Q_{th}$ (see Table 1) are used. The graphs in Fig. 3 show a decreasing similarity of the distribution functions with increasing distances over all temporal resolutions, as the values of $T$ are increasing with increasing distances. Note that the average T-values of the hourly (1H) data in Fig. 3 a are shown as the highest dashed line in 3 b. So the continuity of the whole distribution changes in space, not only the continuity of values of a single quantile. This shows the applicability of interpolation techniques like OK.

## 6 Precipitation amount models

In the following subsections non-parametric and parametric models for precipitation amounts at single sites will be introduced. Before estimating the non-parametric or parametric distributions at each observation gauge, observations smaller than $QV_{th}$ are censored from the sample of each gauge and $QV_{th}$ is subtracted from the values above them to fit to the support of the theoretical distribution functions $[0, \infty)$. $QV_{th}$ varies from gauge to gauge for different temporal resolutions (see Table 1). After estimating the theoretical CDFs, the quantiles $F$ are scaled with $Q_{th}$

$$F_{sc} = F \cdot (1 - Q_{th}) + Q_{th} \qquad (5)$$

and $QV_{th}$ is added to the quantile values. Only the monthly resolution is excluded from the whole scaling procedure, as all monthly rainfall values are used.

### 6.1 Non-parametric models

Non-parametric KDEs for precipitation amount distributions were previously used and are described for daily precipitation amounts in Rajagopalan et al. (1997) and Peel and Wilson (2008). By using this non-parametric method no theoretical distribution needs to be preassigned only a kernel and its bandwidth needs to be chosen. That is why they are assumed to be more flexible. A kernel in this context is a function which is centered over each observation value and is itself a probability density whose variance is controlled by its bandwidth (Bowman and Azzalini, 1997). The probability density function (PDF) or KDE $f(x)$ of every data set is then constructed through a linear superposition of these kernels (Peel and Wilson, 2008), where $n$



is the number of observed values, $K$ is the kernel function, $h$ is the bandwidth of the kernel, $x$ are discrete kernel supporting points and $x_i$ are observed rainfall values:

$$f(x) = \frac{1}{n} \sum_{i=1}^{n} K(x - x_i; h) \tag{6}$$

The estimation of $f(x)$ is performed with an R (R Core Team, 2015) implementation of Wand (2015). However, since our
non-parametric interpolation scheme is based on CDFs and not on PDFs, the CDF is needed. In order to obtain a CDF from the KDEs an integration is required, which is done numerically with the composite trapezoidal rule (e.g., Atkinson, 1989). For numerical reasons CDF values (quantiles) slightly greater than 1 are sometimes obtained, which are simply set to 1 to remain in the correct range.

To model right skewed precipitation amounts with their bounded support on $[0, \infty)$ either an asymmetric kernel like the
Gamma kernel (Chen, 2000) or a symmetric kernel with a prior logarithmic transformation of the values (Rajagopalan et al., 1997) can be used to avoid boundary bias. A boundary bias occurs when kernels with infinite support are used for data with bounded support, as this would lead to a leakage of probability mass (Rajagopalan et al., 1997).

In this work the symmetric Gaussian kernel with a prior transformation of data to logarithms is chosen, as this is an implicit adaptive kernel method with increasing bandwidths for increasing values and therefore alleviates the need to choose variable
bandwidths with skewed data (Lall et al., 1996; Charpentier and Flachaire, 2014). The Gaussian kernel is chosen as it is straightforward and its application is facilitated through several software implementations (Sheather, 2004). The Gaussian kernel $K(t)$ is described in Eq. 7:

$$K(t) = \frac{1}{h\sqrt{2\pi}} \cdot exp\left(\frac{-t^2}{2h^2}\right) \tag{7}$$

If the density of the logarithmically transformed observed values $y = log(x)$ is $f_Y$ and a Gaussian kernel is used for this
density estimation the density estimation $f_X$ of the original values $x$ according to Charpentier and Flachaire (2014) is:

$$f_X(x) = f_Y(log(x))\frac{1}{x} \tag{8}$$

Finally the bandwidth $h$ needs to be chosen, which is commonly indicated as the key step for KDEs (e.g., Bowman, 1984; Harrold et al., 2003; Sheather, 2004; Charpentier and Flachaire, 2014) as a poor bandwidth selection may result in a peakedness or an over smoothing of the density estimation. Due to this great importance of the bandwidth selection the performance of
different selection methods is investigated.

1. The simplest and mostly used selection method is Silverman's rule of thumb (Silverman, 1986), which is defined as

$$h_{opt,SRT} = 0.9 \cdot min\left(s; \frac{q_3 - q_1}{1.349}\right) n^{-1/5} \tag{9}$$

to obtain the optimal kernel bandwidth $h_{opt,SRT}$ with $n$ sample values, where $s$ is the standard deviation and $q_3 - q_1$ is the interquartile range. Silverman's rule of thumb (SRT) is deduced from minimizing an approximation of the mean
integrated squared error between the estimated and the true densities, where the Gaussian distribution is referred to as





the true distribution. This resulted in using the minimum of two measures of dispersion: the standard deviation, which is sensitive to outliers and the interquartile range (Charpentier and Flachaire, 2014).

2. The second method is a plug-in approach developed by Sheather and Jones (1991), which is widely recommended due to its good performance (Jones et al., 1996; Rajagopalan et al., 1997; Sheather, 2004). Instead of using a Gaussian reference distribution it uses a prior non-parametric estimate in the approximation of the mean integrated square error and therefore requires numerical calculation (Charpentier and Flachaire, 2014) to find the optimal bandwidth $h_{opt,SJ}$, which is performed with the R implementation of Wand (2015) within this work.

Instead of minimizing the mean integrated squared error Bowman (1984) recommended minimizing the integrated squared error through a least squares cross validation (LSCV), which is applied using the R package of Duong (2015). Another common cross validation method is the maximum likelihood cross validation (MLCV), where the optimal bandwidth $h$ is obtained by maximizing a pseudo likelihood. Cross validation methods tend to produce small bandwidths and therefore tend to produce peakedness of the density (Rajagopalan et al., 1997; Sheather, 2004; Peel and Wilson, 2008), which we also observed in our applications. This peakedness of the PDF may have been caused by discrete observations, however it could not be fully removed by adding uniformly distributed values between e.g. (0, 0.1) to the observations as described in De Michele et al. (2013). Peakedness of the PDF leads to similarity between CDF and EDF, and is therefore not very useful for further applications, as it makes the advantages of smoothed distributions negligible (see section 3). Due to this deficiency both cross validation methods are not considered in what follows.

## 6.2 Parametric Models

Within the parametric procedure four different parametric distributions are used to model the precipitation amounts of all aggregations in this study. The exponential distribution is a widely used and simple model. Another common model for daily rainfall is the two parameter gamma distribution (Wilks and Wilby, 1999). The mixed exponential distribution was also recommended in (Wilks and Wilby, 1999) and was firstly used for daily precipitation amounts by Woolhiser and Pegram (1979). In addition to these models the Weibull distribution is used, which showed good performance for modeling monthly precipitation amounts in Baden-Württemberg (Beck, 2013). The CDF $F(x)$ and PDF $f(x)$ of each used parametric distribution are listed in the following.

1. For the exponential distribution with the parameter $\lambda$ these functions are:

$$f(x;\lambda) = \lambda e^{-\lambda x} \tag{10}$$

$$F(x;\lambda) = 1 - e^{-\lambda x} \tag{11}$$





2. For the two parameter gamma distribution they are:

$$f(x;\theta,k) = \frac{x^{k-1}e^{-\frac{x}{\theta}}}{\Gamma(k)\theta^k} \tag{12}$$

$$F(x;\theta,k) = \frac{\gamma\left(k,\frac{x}{\theta}\right)}{\Gamma(k)} \tag{13}$$

where $\Gamma$ is the gamma function and $\gamma$ is the incomplete gamma function.

3. For the two parameter Weibull distribution $F(x)$ and $f(x)$ are:

$$f(x;\lambda,k) = \frac{k}{\lambda}\left(\frac{x}{\lambda}\right)^{(k-1)}e^{-(x/\lambda)^k} \tag{14}$$

$$F(x;\lambda,k) = 1 - e^{-(x/\lambda)^k} \tag{15}$$

4. The mixed exponential distribution exhibits the following functions:

$$f(x;\lambda_1,\lambda_2,\alpha) = \alpha\lambda_1 e^{-\lambda_1 x} + (1-\alpha)\lambda_2 e^{-\lambda_2 x} \tag{16}$$

$$F(x;\lambda_1,\lambda_2,\alpha) = 1 - \alpha e^{-\lambda_1 x} - (1-\alpha)e^{-\lambda_2 x} \tag{17}$$

Parametric distributions with more than two parameters are not considered, as this would complicate the regionalization of the distributions due to dependencies among the parameters. For the three parameter mixed exponential distribution the parameter $\alpha$ is fixed for the whole study region (Wilks, 2008), which transforms it into a two parameter distribution.

In order to estimate the optimal parameter sets of the presented parametric distributions for each rainfall gauge and temporal resolution the maximum likelihood method (MLM) and the method of moments (MOM) are applied. The MLM is applied to all mentioned parametric distributions. Within MLM the Likelihood function $ln(L(\vartheta;x_1,...,x_n))$ is maximized by changing the parameter values $\vartheta$ of the respective distribution within a Simplex algorithm. The Likelihood function $L$ consists of the product of the corresponding PDF values of $n$ observations:

$$L(\vartheta;x_1,...,x_n) = \prod_{i=1}^{n} f(x_i|\vartheta) \tag{18}$$

For numerical reasons it is a usual procedure to take the (natural) logarithm of the above product. As the logarithm is a monotonic function the maximum will be obtained with the same parameter combination as before:

$$ln(L(\vartheta;x_1,...,x_n)) = \sum_{i=1}^{n} ln(f(x_i|\vartheta)) \tag{19}$$

In the special case of the mixed exponential distribution the parameter $\alpha$ is varied between 0.01 and 0.5 within the parameter estimation. For each value of $\alpha$ the sum of the log-transformed likelihoods is calculated over all gauges with varying values of the remaining parameters, while the maximum sum defines the parameter set.

To apply MOM to the gamma distribution, the mean $\bar{x}$ and the standard deviation $s_x$ of the sample values ($\theta = \frac{s_x^2}{\bar{x}}$, $k = \frac{\bar{x}^2}{s_x^2}$) are needed. In order to use MOM for the Weibull distribution at first the coefficient of variation $CV$ is determined.

$$CV = \frac{s_x}{\bar{x}} \tag{20}$$





This empirical value of $CV$ is subsequently used to estimate the Weibull parameter $k$ through solving the following Eq. 21 (Cohen, 1965)

$$CV = \frac{\sqrt{\Gamma\left(1 + \frac{2}{k}\right) - \left(\Gamma\left(1 + \frac{1}{k}\right)\right)^2}}{\Gamma\left(1 + \frac{1}{k}\right)} \tag{21}$$

within a Simplex algorithm. The second parameter can subsequently obtained via

$$\lambda = \frac{\bar{x}}{\Gamma\left(1 + \frac{1}{k}\right)}. \tag{22}$$

For the estimation of the mixed exponential distribution parameters MOM is not applied, due to its shortcomings described in Rider (1961). MOM is neither applied to the one parameter exponential distribution, as it would yield the same results as with MLM.

## 7 Non-parametric distributions in a spatial context

In order to establish the basis of the proposed regionalization procedure for non-parametric models and to get a more detailed idea of the spatial relationship of distribution functions, the EDFs of hourly and monthly rainfall intensities of the gauge Stuttgart / Schnarrenberg and its five closest gauges are plotted in Fig. 4. It is therefore not of importance which EDF belongs to which gauge, but rather the relationship the EDFs have with each other. These two graphs show that the order of the EDFs stays quite persistent over different quantiles for both aggregations, as the EDFs do not cross each other very often. In other words, if one gauge exhibits the highest rainfall values for a certain quantile it also exhibits the highest rainfall values for other quantiles and vice versa. The red and purple EDFs on the left graph illustrate this quite nicely.

A more global look at the spatial relation between different EDFs can be obtained with Spearman's rank correlation $\rho_{xy}$ of quantile values of all gauges for different quantile pairs. As we want to investigate the persistence of EDFs for the whole study region, we are only interested in the ranks or rather the order of different quantile values for differing quantiles, which can be done by calculating $\rho_{xy}$.

For the calculation of $\rho_{xy}$ at first the ranks of two datasets $x$ and $y$ need to be determined, which is done by sorting the values and substituting the values with their ranking positions:

$$x = [x_1, x_2, ..., x_n] \quad where \quad x_1 \leq x_2 ... \leq x_n \tag{23}$$

$$Rank(x) = [1, 2, ..., n] \tag{24}$$

To finally obtain the rank correlation $\rho_{xy}$, the correlation $r_{xy}$ is calculated with the ranks of the values instead of the values themselves:

$$\rho_{xy} = \frac{s_{qt}}{s_q \cdot s_t} \tag{25}$$

where $q$ and $t$ are Rank(x) and Rank(y), $s_{qt}$ is the covariance of $q$ and $t$, $s_q$ and $s_t$ are the standard deviations of $s$ and $t$. In our case the two datasets $x$ and $y$ represent quantile values of two different pairs of quantiles over all gauges in the study region.





These pairwise rank correlations of quantile values of all gauge pairs are calculated starting from $Q_{th}$ until 1 in 0.001 steps for sub-daily aggregations and in 0.005 steps for aggregations greater or equal to one day. This procedure is repeated until the rank correlation of every quantile with every other quantile is obtained. Finally the mean values of the rank correlation values belonging to each quantile are calculated (see the dotted gray lines in Fig. 5). The greatest mean rank correlation is

indicated with a red cross in this figure which also defines the control quantile ($Q_c$) with the highest mean rank correlation. Rank correlations of $Q_c$ with the remaining quantiles lead to the dashed lines in Fig. 5.

Fig. 5 demonstrates, that most of the rank correlation values are greater than 0.85, which indicates a persistence of quantile values over a great interval of quantiles as well as over the whole study region for hourly through monthly data. Lower correlation values can be observed for the highest and lowest quantiles, which indicates a non persistent behavior for these

quantiles. This behavior is similar for all temporal resolutions. Therefore, quantile values of $Q_c$ can be used to set up the interpolation weights. Applying these weights to the remaining quantiles from $Q_{th}$ until 1 should lead to good regionalization results for non-parametric CDFs.

In Table 2 the control quantiles $Q_c$ with the highest mean correlations are summarized for all temporal resolutions. As the precipitation mechanisms are different in summer and winter in Baden-Württemberg, the rainfall data sets are also analyzed

separately for summer (from May to August) and winter (from September to April). $Q_c$ is mostly close to the center of the considered quantile ranges, which are also shown in Table 2. Nevertheless, it is worth noting the strong similarity of winter and summer control quantiles $Q_c$. The proposed procedure to interpolate non-parametric distribution functions using the same interpolation weights for different quantiles seems feasible as the persistence of the order for quantile values of spatially distributed rain gauges is evident. Only values of very high and low quantiles show a non-persistent behavior. Therefore,

quality measures will be introduced, which focus on the difference of these values.

## 8   Regionalizing of precipitation amount models

Until now we have introduced the different methods to model precipitation amounts at point locations, where observations of rainfall are available. Additionally, we have investigated the basis for the proposed non-parametric interpolation procedure. In the following, the regionalization of point models in order to obtain precipitation amount models at ungauged locations will be

described. The regionalization method used for this purpose is OK.

OK will be introduced first in 8.1 and then the regionalization of parametric precipitation amount models will be outlined briefly in subsection 8.2. The newly developed regionalization procedure for the non-parametric models will be explained in more detail in subsection 8.3, as it differs fundamentally from the common regionalization of parametric models.

### 8.1   Regionalization technique

In the following only a short overview of OK will be given. For further information the interested reader is referred to the common geostatistical literature like Kitanidis (1997). The first step of OK is fitting a theoretical variogram model to the empirical variogram. The variogram describes the development of the spatial dependence over increasing distances of the





regarded variable. The empirical variogram $\gamma_e(h)$ is calculated using Eq. 26

$$\gamma_e(h) = \frac{1}{2n(h)} \sum_{i=1}^{n(h)} (z(x_i) - z(x_i + h))^2 \tag{26}$$

where $n(h)$ is the number of gauge pairs for distance $h$, $x_i$ represents the position of a gauge $i$ and $z(x_i)$ is the variable value at gauge $i$. As the distances between rainfall gauges never provide a continuous set of distances, the $h$ in Eq. 26 represents

different distance intervals. For following applications the width of the interval of $h$ is 10 km. For the theoretical variogram $\gamma_t(h)$ one single model out of the following four is chosen based on the least squares criterion. The $s$ parameters represent the sills, the $r$ parameters the ranges of the variograms.

1.  Gauss model:

$$\gamma_t(h) = s_1 \left( 1 - e^{-\frac{h^2}{r_1^2}} \right) \tag{27}$$

2.  Spherical model:

$$\gamma_t(h) = s_2 \left( 1.5 \frac{h}{r_2} - 0.5 \left( \frac{h}{r_2} \right)^3 \right) \tag{28}$$

3.  Exponential model:

$$\gamma_t(h) = s_3 \left( 1 - e^{-\frac{h}{r_3}} \right) \tag{29}$$

4.  Matern model (Pardo-Iguzquiza and Chica-Olmo (2008), $K_v$ is the modified bessel function of second kind):

$$\gamma_t(h) = s_4 \left( 1 - \frac{1}{2^{v-1}\Gamma(v)} \left( \frac{h}{r_4} \right)^v K_v \left( \frac{h}{r_4} \right) \right) \tag{30}$$

The next step within OK is solving the corresponding equation system to estimate an interpolated value at an unobserved location $x_0$:

$$\sum_{j=1}^{n} \phi_j \gamma_t(x_i - x_j) + \mu = \gamma_t(x_i - x_0) \qquad i = 1,...,n, \tag{31}$$

$$\sum_{j=1}^{n} \phi_j = 1. \tag{32}$$

where $n$ is the number of gauges included in the interpolation (10 within this work) and $\mu$ is the Lagrange multiplier.

## 8.2  Parametric Models

As already outlined in the introduction, either the parameters (Kleiber et al., 2012) or the moments (Haberlandt, 1998; Wilks, 2008) of parametric distributions can be interpolated to regionalize parametric models. Within this work the moments are





interpolated, when MOM is used for fitting the parametric distributions. If MLM is used, the parameters are interpolated. Since only rainfall values above $QV_{th}$ (see Table 1) are used, $QV_{th}$ also needs to be interpolated within the parametric approach for the different temporal resolutions. The resulting regionalized $QV_{th}$ then serve as anchor points for the parametric CDFs, whose shape is determined by the regionalized parameters of the respective distribution.

## 8.3 Non-parametric models

Kernel smoothed distribution functions do not provide a parameter that can be interpolated, so a procedure other than for parametric distributions needs to be applied. By analyzing the spatial relation of rainfall EDFs in section 7, a persistent order of quantile values over a wide range of quantiles is observed. Therefore, the interpolation weights of quantile values for the control quantile $Q_c$ (see Table 2) can be applied to the remaining quantiles.

For all gauges the quantile values $QV_c$ of the control quantile $Q_c$ are estimated with the inverse of the gauge wise numerically integrated non-parametric CDF $F_{np}$:

$$QV_c = F_{np}^{-1}(Q_c) \tag{33}$$

With these $QV_c$ at the observation points, the interpolation weights $\phi_j$ for the target locations are estimated with OK (see Eq. 31). Then these weights are applied to the quantile values of quantiles between $Q_{th}$ and 1 in 0.0001 steps and finally the remaining quantile values are linearly interpolated to receive a continuous CDF for all target locations. In order to assure a monotonically increasing CDF only positive interpolation weights are allowed. This makes the use of OK problematic. It can only be used if the equation system (see Eq. 31) is solved with positive weights, which leads to an additional constraint:

$$\phi_j \geq 0 \qquad j = 1, ..., n. \tag{34}$$

Considering this additional constraint the OK equation system is solved with a SCIPY implementation (Jones et al., 2001) of a FORTRAN algorithm by Lawson and Hanson (1987), which solves the Karush-Kuhn-Tucker conditions for the non-negative least squares problem. In the following this kriging procedure will be called positive kriging (PK). Another way to solve this extended optimization problem with an application of the Lagrange method is presented in Szidarovszky et al. (1987).

The persistence of quantile values (inverse of the quantiles) described in section 7 also implies the persistence of quantiles. The interpolation of quantiles for discrete rainfall values would therefore also be an option. However, this would complicate the regionalization as not only monotonicity needs to be preserved, but also the value range of quantiles from 0 to 1.

## 9 Regionalization example

As already mentioned in the introduction, two main possibilities to obtain precipitation amount distributions at ungauged locations exist. In the following, these possibilities will be compared with each other. In order to assure equal interpolation weights $\epsilon_i$ of the control gauges $i$ for both interpolation possibilities, a simple inverse distance weighting (IDW) is used as





interpolation technique in this example, which is based on the following Eq. 35:

$$\epsilon_i = \frac{\frac{1}{d_i^2}}{\sum\limits_{i=1}^{30} \frac{1}{d_i^2}} \tag{35}$$

where $d_i$ is the distance between control gauge $i$ and the respective target gauge.

The first interpolation method is the interpolation of rainfall values for every time step to the target location, followed by

an estimation of the distribution function with the interpolated values ($values_{idw}$). The second approach is first fitting a non-parametric distribution function to all control locations, which is followed by an interpolation of these distribution functions to the target location ($cdf_{idw}$). In the following example the non-parametric KDE using SRT for the bandwidth selection is applied for estimating the distribution functions at the control gauges.

Although it is commonly accepted to follow the $cdf_{idw}$ approach to obtain precipitation amount distributions at ungauged

locations for stochastic rainfall models, we still want to illustrate the deficiencies of the $values_{idw}$ method to motivate the $cdf_{idw}$ approach empirically. Additionally, the resulting estimation errors also appear when rainfall values are interpolated without considering the CDF explicitly. For example the use of interpolated rainfall values for hydrological models may introduce a bias in the discharge estimation caused by the poor interpolation results.

In our example the distribution of daily rainfall values (1D) for the gauge Esslingen / Neckar is estimated from rainfall values

of 30 neighboring gauges (see Fig. 6 a). In Fig. 6 b and c, parts of the distribution functions resulting from both methods and the original EDF are shown. Clear disadvantages of the $values_{idw}$ method are the overestimation of days with rainfall and thus an underestimation of the probability of no rainfall (Fig. 6 b) and a clear underestimation of the CDF for higher quantiles (Fig. 6 c).

As the $cdf_{idw}$ method does not provide rainfall values automatically, which are needed to calculate basic statistical measures,

random rainfall values are generated with the inverse of the interpolated non-parametric CDF. The number of these random values is equivalent to the number of observed daily rainfall values of the validation gauge. In Table 3 basic statistics of precipitation amounts are listed for both methods and the observations. Looking at the mean values of all rainfall ($\overline{x}$) values, the $values_{idw}$ method seems to reproduce this statistic very well. Considering the other statistics in Table 3 and Fig. 6 this is most probably caused by two disadvantages of this method: an overestimation of days with small rainfall amounts (see $P_0$) and

a simultaneous underestimation of higher rainfall intensities (see $\overline{x}_{>0}$ and $max$). This argument is reaffirmed by the smaller standard deviation of $values_{idw}$ and the illustrations of the precipitation amount distributions in Fig. 6. The $cdf_{idw}$ method mainly provides better results summarizing the listed statistics. Only a tendency of overestimating high rainfall intensities can be observed.

As the $values_{idw}$ method has great problems in reproducing probabilities of zero rainfall and the course of the distribution

function, this method is not recommended to be used with rainfall over a great range of aggregations. For higher aggregations these disadvantages may have no great effect, but for smaller aggregations with a greater skewness the problems might even increase. This would lead to a more pronounced under estimation of high quantile values, which are mostly the decisive ones for subsequent applications.



As the $cdf_{idw}$ method exhibits better results concerning the basic rainfall volume statistics, it seems to be the better choice for the purpose of interpolating precipitation amount models, so it will be adopted in the sequel with OK as interpolation technique. However, before we come to the results, the use of daily values for sub-daily statistics will be investigated in the following section.

## 10 Dependence of sub-daily on daily values

As the high resolution rain gauge monitoring network in the study area is quite sparse and the corresponding time series are often incomplete, it would be useful to include more dense and complete secondary information in the interpolation of the sub-daily distributions. Therefore the applicability of daily values to improve their interpolation is investigated, as the daily monitoring network exhibits a higher density. A simple disaggregation strategy (rescaled nearest neighbor) of Bárdossy and Pegram (2016) is applied to all days to obtain distributions of sub-daily resolutions at the locations of the daily gauges.

The procedure to incorporate daily values in the interpolation of sub-daily values should be the following:

1. Choose a daily gauge and assign the rainfall values of the closest (concerning horizontal distance) high resolution gauge to it.

2. Aggregate the hourly values to daily values $p_{hourly}(t)$ and calculate a scaling factor for every day $t$ with the values of the daily gauge $p_{daily}(t)$ :

$$sc(t) = \frac{p_{daily}(t)}{p_{hourly}(t)} \tag{36}$$

3. Multiply all hourly values of the nearest gauge with this scaling factor (varying from day to day).

4. Repeat steps 1. to 3. for all daily gauges.

5. Calculate the sub-daily statistic of interest from these scaled values at every daily gauge and incorporate them in the interpolation procedure.

To estimate the applicability of this procedure a cross validation is applied based on the high resolution gauges only, which are used as daily gauges one after another. The resulting sub-daily statistics of scaled values for these pseudo daily gauges are compared to their original sub-daily values by calculating the mean squared errors over all gauges. The scaled nearest neighbor values are compared to nearest neighbor values and to interpolated rainfall values. The interpolation is done by OK with ten neighbors using a single variogram model. During the cross validation a nearest neighbor gauge is defined as the gauge with the closest distance and at least 50 % of data overlapping. For the interpolation then again only the overlapping period is chosen.

In Fig. 7 the results are shown for quantile values, but the standard deviation, the mean values and $QV_{th}$ were also investigated. The cross validation of the different statistical variables are very similar. For all of them the scaled nearest neighbor values (NNS) lead to the best results in summer and winter. Therefore daily gauges seem to be useful for the interpolation of sub-daily non-parametric and parametric models.





For the incorporation of daily values within the regionalization of parametric and non-parametric sub-daily distributions a special regionalization technique is not needed. The rescaling method (NNS) is applied to all available daily gauges with a minor change. If for a certain day no hourly values are available for the closest gauge the next closest gauge is used for the rescaling of that certain day in order to increase the sub-daily sample size at the daily gauge. After obtaining the sub-daily
values at the daily gauges, they are simply treated as additional control points for the regionalization.

## 11   Performance

This section is divided into three parts. In the first part 11.1 the quality measures will be introduced, in the second 11.2 the performance of the precipitation amount models for point wise estimations are compared for all temporal resolutions. In 11.3 the regionalization of the precipitation amount models is addressed. The precipitation amount models are fitted and
regionalized separately for winter (from September to April) and summer (from May to August) months, as the rain-producing weather processes are different in these two seasons.

### 11.1   Quality measures

The validation of the precipitation amount models at point locations and their regionalization is evaluated with two different quality measures. These quality measures need to be measures considering the CDF and not the PDF as the interpolation of the
non-parametric distributions only provides CDFs for ungauged locations.

The most common goodness of fit test to estimate the quality of fitted distributions is the Kolmogorov–Smirnov test. As distributions of precipitation amounts are positively skewed the greatest part of the values are small or medium values, which leads to the highest gradient of the CDF for these values. Therefore, a greater difference of the corresponding CDF quantiles would be more likely and would govern the Kolmogorov–Smirnov test. However, these medium values are less important than
the greater precipitation amounts for most of the precipitation model applications.

For this reason the Cramér–von Mises criterion as a more integral measure and a Lorenz-curve based measure - which allows for conclusions about the representation of the water volume - are used. The Cramér - von Mises criterion $W^2$ for single samples is (Stephens, 1974):

$$W^2 = \frac{1}{12n} \sum_{i=1}^{n} \left( \frac{2i-1}{2n} - F(x_i) \right)^2 \tag{37}$$

where $F(x_i)$ represents the theoretical distribution (non-parametric or parametric) of the observed values $x_i$ in ascending order. For sub monthly resolutions the Cramér - von Mises criterion is slightly modified, as only quantiles above $Q_{th}$ (see Table 1) are used:

$$W^2 = \frac{1}{12n} \sum_{i=1}^{n} \left( \left( \frac{2i-1}{2n} \cdot (1 - Q_{th}) + Q_{th} \right) - F(x_i) \right)^2 \tag{38}$$

As already mentioned in section 7, a quality measure is needed, which describes the representation of high quantiles. For
Lorenz-curves, high vertical differences are supposed to appear more frequently for high quantiles as the slope increases with





increasing probabilities for rainfall intensities. Therefore a measure respecting the vertical differences of the Lorenz-curves is suitable. In section 4 the estimation of the Lorenz-curve with observed rainfall values was described. However, the Lorenz-curve $L(F(x))$ can also be estimated from the theoretical CDF $F(x)$, which is a preferable approach, as random rainfall values don't need to be generated from the CDF previous to the Lorenz-curve estimation:

$$L(F(x)) = \frac{\int_0^F x(F)dF}{\int_0^1 x(F)dF} \tag{39}$$

where $x(F)$ is the gauge wise quantile function (the inverse of the CDF). The integrals of the quantile functions are estimated numerically as the non-parametrically estimated distribution functions are not analytically invertible. The Lorenz-curve criterion $L_d$ used here is the squared difference of the observed $L(F_n(x))$ and modeled Lorenz-curve $L(F(x))$:

$$L_d = \sum_{i=1}^{n} (L(F_n(x)) - L(F(x)))^2 \tag{40}$$

The differences of the Lorenz-curves are only estimated for values greater than $QV_{th}$ (see Table 1). Within the validation of the regionalization only values above the highest $QV_{th}$ among the observed and differently regionalized values for each gauge are evaluated, as they may differ for the different techniques.

## 11.2 Point models

To determine an overall performance ranking for the remaining models, at first the arithmetic mean and the median over the number of gauges of both measures of quality - the Cramér–von Mises criterion $W^2$ and the Lorenz-curve criterion $L_d$ - are calculated for each precipitation amount model. This leads to four different measures, which are shown for hourly values of the winter season in Table 4. Note that mean values reflect the robustness and median values represent a good average performance of one precipitation model for the whole study region.

To combine the four statistics (mean and median of $W^2$ and $L_d$ respectively) in one single performance measure every value in Table 4 is then divided by the smallest (best) value (bold numbers) of its corresponding quality measure, indicating the relative performance with respect to the best model. This leads to one number for each statistic and precipitation model starting from 1 for the best performing model of each statistic. The bigger this number the worse its relative performance. These four numbers are then combined by adding them together, which results in a single number for each precipitation amount model to define the performance ranking for each temporal resolution. A ranking number of 4 is the lowest possible number and implies that the related model shows the best performance for all four quality measures. In Table 5 the ranking numbers for all temporal resolutions and both seasons are shown.

With the ranking numbers the best performing precipitation amount model is estimated for each season and temporal resolution. In Table 6 the best parametric and non-parametric methods are presented for each resolution and it is stated whether the non-parametric or parametric method performs better. Among the non-parametric methods (NP) Silverman's rule of thumb (SRT) and the plug-in approach of Sheather and Jones (1991) (SJ) show very similar results, especially in the winter season. The mixed exponential distribution with a MLM parameter estimation (Mixed-Exp-MLM) leads to the best results among the



parametric methods (P) for daily and sub-daily resolutions. For temporal resolutions greater 1D the Weibull distribution with a MOM parameter estimation (Weibull-MOM) leads to the best results. The best performance of the Weibull distribution for monthly values coincides with the results of Beck (2013) for the same study region.

The performance ranking of the different methods is quite similar in winter and summer. The non-parametric methods always lead to better performances concerning the Cramér - von Mises criterion $W^2$. The parametric estimations mostly lead to better results regarding the Lorenz-curve criterion $L_d$. Figure 8 may provide an explanation for the differences in performance regarding these two quality measures. The graphs show the CDFs and Lorenz-curves for the hourly (1H) and 12 hourly (12H) resolution for a chosen gauge. For the hourly resolution the non-parametric SRT method leads to better results for both measures. An equally good performance regarding the $W^2$ for the parametric and non-parametric method can be observed for the 12 hourly resolution. However, the non-parametric method performs worse regarding the $L_d$ measure, as it overestimates the water volume represented by the higher quantiles. The reason can already be observed in the CDF, where the non-parametric method systematically overestimates the values of high quantiles. The parametric method can lead to both to over- and underestimations. This influences the $W^2$ criterion in the same way as a constant overestimation (see squared differences in Eq. 37), but it seems to lead to better results regarding the $L_d$ criterion.

Parameter estimation through MOM in combination with the Weibull distribution performs better for higher aggregations, which exhibit more symmetric distributions. For daily and sub-daily aggregations, MLM parameter estimation in combination with the mixed exponential distribution leads to better results.

The overall performance is best with the mixed exponential distribution for temporal resolutions between two hours (2H) and one day (1D) in both seasons. For five daily (5D) resolutions, the Weibull distribution exhibits the best overall performance in both seasons. For the hourly distribution (1H), the non-parametric models show the best overall performance in both seasons. Only for the monthly distribution (M) the best performing methods differ between the two seasons. In the summer season the Weibull distribution shows the best results and in the winter season the non-parametric models perform best.

## 11.3 Regionalization

The development of the two-sample Cramér–von Mises criterion $T$ over distance in section 5 mostly indicates a spatial dependence which may be modeled reasonably via kriging. Additionally, kriging techniques were already applied successfully by others for the interpolation of rainfall values over different temporal scales (e.g., Tobin et al., 2011; Lloyd, 2005).

In order to estimate the quality of the regionalized precipitation amount models a 2-fold cross validation (split sampling) is used. Two equally sized samples of observation points are randomly generated (Fig. 9). The most simple regionalization method is using the estimates of the nearest neighbor (NN) of the calibration set, which are therefore used as benchmarks for the quality of the regionalization procedure. Additionally the daily rescaled nearest neighbors (NNS) are used as benchmark. In this case all daily gauges are used for the rescaling except for the daily observations at the locations of the respective validation sample.

Following the results of the point-wise estimation in the previous section only the Weibull-MOM and the Mixed-Exp-MLM models among the parametric models are investigated for the regionalization, as they show good performance for differing





aggregations. They are both investigated for all aggregations to test the difference of interpolating moments or parameters, except for the monthly aggregation, for which only the Weibull distribution is investigated. In order to regionalize the Weibull-MOM model the mean and standard deviation are spatially interpolated, for the regionalization of the Mixed-Exp-MLM model its parameters $\lambda_1$ and $\lambda_2$ are interpolated while its parameter $\alpha$ is constant for the whole study region.

5    As the two non-parametric approaches SRT and SJ show very similar results during the point wise estimation only the SRT approach is interpolated. For the regionalization of the non-parametric models $QV_c$ (see Table 2 and Eq. 33) values are used to estimate the interpolation weights, which are further applied to the remaining quantiles.

Following the conclusions in section 10, daily gauges can be used to set up distribution functions for sub-daily values with a scaled nearest neighbor approach (NNS). Therefore, their incorporation does not require a special interpolation method as they can be used simply as additional supporting points.

### 11.3.1  Variogram estimation

The first step during the regionalization procedure is the estimation of the theoretical variograms. As described in section 8.1 single theoretical variograms are estimated for all interpolation variables. The interpolation variables of the three precipitation amount models for which theoretical variograms need to be estimated for the two seasons and eight temporal resolutions are:

15    1. P-Mixed-Exp-MLM: $\lambda_1$, $\lambda_1$

2. P-Weibull-MOM: mean, standard deviation

3. NP-SRT: $QV_c$ values (see Table 2 and Eq. 33)

During the estimation of the parameters of the Weibull distribution with MOM, $QV_{th}$ is subtracted from the rainfall values prior to the estimation of the mean and the standard deviation. As the mean of these values show lower spatial dependencies than the mean of the censored values without subtraction, $QV_{th}$ is added to the mean values of the parameter estimation before the regionalization. After the regionalization they are subtracted again to determine the parameters of the Weibull distribution. Variogram models are also fitted to $QV_{th}$ as the corresponding values serve as starting points for the parametric models at the ungauged locations.

### 11.3.2  Precipitation amount models

25    The regionalization of the precipitation amount models is evaluated with the same quality measures as the point wise estimation, the Cramér–von Mises criterion $W^2$ and the Lorenz-curve criterion $L_d$. For both seasons a 2-fold cross validation is applied.

The investigated interpolation approaches for the parametric distributions are:

1. OK - MOM: OK of the Weibull distribution, fitted with MOM.

2. OK - MLM: OK of the mixed exponential distribution, fitted with MLM.





3. OK - MOM Daily: OK of the Weibull distribution including scaled NNS values of daily gauges (only for sub-daily aggregations).

4. OK - MLM Daily: OK of the mixed exponential distribution including scaled NNS values of daily gauges (only for sub-daily aggregations).

The interpolation approaches for the non-parametric models are:

1. PK - NP: PK of the non-parametric models, which are estimated using SRT.

2. PK - NP Daily: PK of the non-parametric models including scaled NNS values of daily gauges (only for sub-daily aggregations).

In Fig. 10, parts of the interpolation procedure for PK - NP are shown for the daily aggregation, where the non-parametric
$QV_c$ at the calibration gauges and three interpolation fields are shown.

In Table 7 and Table 8 the performance ranking numbers of the regionalized precipitation amount models are summarized for the winter season and for the summer season respectively. The differences between the two cross validation samples are quite small, so the performances are not just resulting from the positioning of the gauges in the samples but from the interpolation approaches. Among the parametric methods the MOM approaches mostly perform better than the MLM approaches for
aggregations greater equal 2H during the winter season. In the summer season the MOM approaches perform mostly worse than the MLM approaches for aggregations smaller 6H and vice versa for higher aggregations. Interpolating moments therefore seems to be more robust than interpolating parameters of distributions as the performance ranking changed in favor of the MOM approaches comparing to the point wise results (see Table 6). Only for stronger skewed distributions in the summer and smaller aggregations the MLM approach still outperforms the MOM approach.

Comparing the non-parametric interpolation approaches with the parametric interpolation approaches shows that the non-parametric approach performs best for hourly (1H) and two hourly values (2H) for both calibration samples and for calibration sample 1 with three hourly values (3H) in the winter season. This seems to indicate a more robust non-parametric interpolation method for the winter season as the performance ranking changed in favor of it compared to the point wise estimation. In the summer season the non-parametric methods only perform best for the hourly resolution (1H), which is similar to the results of
the point wise estimation.

It is obvious that using scaled values of the daily gauges is very beneficial as approaches incorporating these values almost always include the best performing method except for the 12H aggregation in the summer season.

As benchmark the interpolation results are also shown for parametric and non-parametric estimates of nearest neighbors (NN) and additionally using scaled daily gauges for sub-daily aggregations (NNS). Among the benchmark methods the NNS
approaches perform better than the simpler NN approaches for the sub-daily aggregations, except for the twelve hourly (12H) resolution in summer. Since the best interpolation approach almost always - with only two exceptions - performs better than the best nearest neighbor approach the regionalization of distributions seems to be worthwhile.





## 12  Conclusions

Comparing different modeling schemes for precipitation amounts at point locations (see Table 6) over different temporal resolutions has revealed several findings. The non-parametric estimates only perform better for the hourly resolution in both seasons and for monthly distributions in the winter season. The non-parametric estimates especially have problems in reproducing the

volume correctly, as they seem to have difficulties with high quantiles. Causes for this deficiency could be the numeric interpolation or the small number of rainfall values at high quantiles. For temporal resolutions between two hours and a month the parametric distributions outperform the non-parametric estimates. Among the parametric methods the mixed exponential distribution performs better for sub-daily and daily aggregations, whereas the Weibull distribution has the advantage for higher aggregations.

The regionalization of the precipitation amount models showed (see Table 7 and 8) that the proposed interpolation scheme for non-parametric distributions is useful as it does not worsen its performance ranking compared to the estimation at point locations. Rather, it appears to be a robust interpolation scheme as it more often outperforms the parametric schemes comparing point wise estimation and regionalization. Among the parametric methods the interpolation of moments turned out to be more robust than the interpolation of parameters.

As auxiliary variables, the use of daily gauges for sub-daily resolutions is very beneficial, which was suggested by a data analysis and is proven by the evaluation of the regionalization.

In general the regionalization of distributions seems to be worthwhile as it nearly always performs better than the nearest neighbor (horizontal distance) approaches, which would be the most simple estimate. As lower rainfall values were excluded in this study due to their minor importance and measurement errors, the results are not directly comparable to those of most of

the other publications within this research field.

The difficulty of non-parametric distributions in representing water volumes may be reduced by using the Epanechnikov kernel with finite support as proposed by Rajagopalan et al. (1997). Additionally, ways of incorporating elevation within the regionalization of non-parametric distributions needs to be tested. Regarding the parametric distributions, Chen and Brissette (2014) and Li et al. (2012) recommended Pareto type distributions instead of exponential type distributions, which could also

be tested in the future. Finally, the non-parametric interpolation approach could also be applied to parametric or empirical distributions and should be tested for various study regions.

## 13  Competing interests

The authors declare that they have no conflict of interest.

## 14  Data availability

The used sub-daily precipitation data sets were obtained from the LUBW during various research projects and are not available to the public as far as the authors know. Therefore, they can not be provided by the authors. The daily data set was downloaded



from the Webwerdis homepage (http://www.dwd.de/DE/leistungen/webwerdis/webwerdis.html) of the DWD, where a personal account is required. However, to the knowledge of the authors any academic researcher can apply for a personal account and some of the used daily and sub-daily values also seem to be available at the homepage of the DWD Climate Data Center (http://www.dwd.de/DE/klimaumwelt/cdc/cdc_node.html).

5  *Acknowledgements.* The work of many people developing different libraries of the PYTHON programming language (Jones et al., 2001; van der Walt et al., 2011; McKinney, 2010) made the work on the presented research much more comfortable and helped a lot to illustrate (Hunter, 2007) its findings. The authors appreciate the valuable comments on the manuscript by Geoffrey Pegram (University of KwaZulu-Natal, Durban, South Africa). The authors also want to thank the German Federal Ministry of Education and Research (BMBF) for funding this work through the funding measure *Smart and Multifunctional Infrastructural Systems for Sustainable Water Supply, Sanitation and*
10  *Stormwater Management (INIS)*. Furthermore the authors would like to thank the Environmental Agency of Baden-Württemberg (LUBW) and the German Meteorological Service (DWD) for the provision of rainfall data.





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





**Table 1.** Basic rainfall information of the study region for different aggregations (agg): $P_0$ is the probability of 0 mm rainfall, $Q_{th}$ stands for the defined quantile thresholds or threshold ranges and $QV_{th}$ represents the corresponding quantile values (rainfall) for the defined $Q_{th}$.

| agg | $P_0$ (-) | $Q_{th}$ (-) | $QV_{th}$ (mm) |
|-----|-----------|--------------|----------------|
| 1H | 0.82 - 0.93 | 0.95 | 0.2 - 1.6 |
| 2H | 0.76 - 0.9 | 0.93 | 0.3 - 2.3 |
| 3H | 0.71 - 0.87 | 0.92 | 0.4 - 3.1 |
| 6H | 0.61 - 0.81 | 0.9 | 0.7 - 5.1 |
| 12H | 0.46 - 0.72 | 0.86 | 1.2 - 7.7 |
| 1D | 0.38 - 0.6 | 0.72 | 1.0 - 6.4 |
| 5D | 0.1 - 0.22 | 0.29 | 1.0 - 7.2 |
| M | 0.0 - 0.02 | 0.0 - 0.02 | 0 |

**Table 2.** Control quantiles ($Q_c$) which exhibit the highest mean pair wise rank correlations with other quantiles. They are shown for different temporal aggregations (agg) and separately for summer and winter. Additionaly the (center) quantile in the middle of the investigated quantile range is shown.

| agg | season | | center quantile |
|-----|--------|--------|-----------------|
| | winter | summer | |
| 1H | 0.977 | 0.979 | 0.975 |
| 2H | 0.963 | 0.967 | 0.965 |
| 3H | 0.959 | 0.966 | 0.96 |
| 6H | 0.949 | 0.953 | 0.95 |
| 12H | 0.924 | 0.922 | 0.93 |
| 1D | 0.835 | 0.865 | 0.86 |
| 5D | 0.615 | 0.575 | 0.645 |
| M | 0.545 | 0.46 | 0.5 |





**Table 3.** Regionalization example: Basic daily rainfall statistics of the observed values at the validation gauge ($data$), the interpolated rainfall values ($values_{idw}$), randomly sampled rainfall values of the interpolated non-parametric distribution function ($cdf_{idw}$) and the respective ranges of the calibration gauges. The rainfall statistics are the arithmetic mean ($\overline{x}$), the standard deviation ($s_x$) of all rainfall values, the arithmetic mean ($\overline{x}_{>0}$) of non zero values, the probability of zero rainfall $P_0$ and the maximum value ($max$).

|  | $data$ | $values_{idw}$ | $cdf_{idw}$ | $range\,calibration\,set$ |
|---|---|---|---|---|
| $\overline{x}$ | 2.18 | 2.17 | 2.27 | 1.77 - 3.18 |
| $s_x$ | 4.56 | 4.04 | 4.93 | 3.88 - 6.47 |
| $\overline{x}_{>0}$ | 4.39 | 2.97 | 4.20 | 3.73 - 4.47 |
| $P_0$ | 0.50 | 0.27 | 0.46 | 0.46 - 0.54 |
| $max$ | 56.0 | 49.12 | 62.29 | 42.5 - 102.3 |

**Table 4.** Mean and median of the two quality measures $W^2$ and $L_d$ for the eight precipitation amount models over the study region for hourly values (1H) in the winter season. The bold numbers indicate the lowest (best) value of the corresponding measure.

|  | $W^2$ | | $L_d$ | |
|---|---|---|---|---|
|  | mean | median | mean | median |
| P-Weibull-MLM | 0.001583 | 0.001196 | 0.05294 | 0.03385 |
| P-Gamma-MLM | 0.002638 | 0.00216 | 0.09383 | 0.06477 |
| P-Exp-MLM | 0.00972 | 0.008105 | 0.2712 | 0.2293 |
| P-Mixed-Exp-MLM | 0.0007976 | 0.0004327 | 0.03094 | 0.01646 |
| P-Weibull-MOM | 0.01416 | 0.008296 | 0.06319 | 0.02815 |
| P-Gamma-MOM | 0.03067 | 0.01882 | 0.1378 | 0.0675 |
| NP-SJ | **0.0003486** | **0.0001954** | 0.009708 | **0.005123** |
| NP-SRT | 0.0003753 | 0.0001994 | **0.009673** | 0.005147 |



**Table 5.** Performance ranking numbers of the precipitation amount models for the point wise estimation. The underlined numbers indicate the best parametric (P) and non-parametric (NP) models. The bold numbers indicate the overall best model.

| | 1H | 2H | 3H | 6H | 12H | 1D | 5D | M |
|---|---|---|---|---|---|---|---|---|
| **Winter Season** | | | | | | | | |
| P-Weibull-MLM | 22.74 | 13.04 | 10.81 | 7.15 | 5.45 | 8.38 | 7.26 | 31.49 |
| P-Gamma-MLM | 40.96 | 26.21 | 23.77 | 15.17 | 8.6 | 13.24 | 10.39 | 619.05 |
| P-Exp-MLM | 142.16 | 114.17 | 120.39 | 95.52 | 48.33 | 48.57 | 16.19 | 4744.19 |
| P-Mixed-Exp-MLM | 10.91 | **5.62** | **5.35** | **5.16** | **4.83** | **5.11** | 19.81 | 4746.51 |
| P-Weibull-MOM | 95.1 | 48.09 | 35.45 | 21.62 | 13.08 | 14.05 | **5.9** | 11.4 |
| P-Gamma-MOM | 211.72 | 112.95 | 80.16 | 44.01 | 21.72 | 24.63 | 6.81 | 243.85 |
| NP-SJ | **4** | 7.35 | 18.04 | 45.73 | 43.34 | 31.73 | 12.09 | 10.11 |
| NP-SRT | 4.1 | 7.46 | 18.45 | 44.3 | 40.37 | 34 | 12.5 | **7.37** |
| **Summer Season** | | | | | | | | |
| P-Weibull-MLM | 28.95 | 13.56 | 12.13 | 8.81 | 6.23 | 6.31 | 8.63 | 27 |
| P-Gamma-MLM | 66.05 | 35.66 | 31.95 | 18.38 | 9.08 | 8.68 | 13.87 | 421.39 |
| P-Exp-MLM | 291.57 | 191.99 | 181.42 | 97.26 | 34.39 | 28.48 | 30.79 | 5555.11 |
| P-Mixed-Exp-MLM | 14.38 | **5.93** | **5.31** | **4.69** | **4.72** | **4.71** | 36.06 | 5555.11 |
| P-Weibull-MOM | 98.96 | 41.81 | 28.86 | 17.22 | 10.42 | 8.54 | **5.46** | **7.14** |
| P-Gamma-MOM | 294.99 | 126.61 | 77.35 | 35.78 | 16.9 | 13.66 | 7.21 | 317.29 |
| NP-SJ | 4.47 | 9.93 | 22.73 | 52.92 | 50.11 | 35.1 | 13.03 | 9.73 |
| NP-SRT | **4** | 8.44 | 20.08 | 41.91 | 43.51 | 32.63 | 14.05 | 7.7 |





**Table 6.** Best performing non-parametric (NP) and parametric (P) models for all temporal resolutions and seasons for the point wise estimation. Bold letters indicate the better performing method comparing the non-parametric and parametric estimations. The numbers in parenthesis refer to the ranking numbers in Table 5.

|  | Winter Season | | Summer Season | |
|  | NP | P | NP | P |
|---|---|---|---|---|
| 1H | **SJ** (4) | Mixed-Exp-MLM (10.91) | **SRT** (4) | Mixed-Exp-MLM (14.38) |
| 2H | SJ (7.35) | **Mixed-Exp-MLM** (5.62) | SRT (8.44) | **Mixed-Exp-MLM** (5.93) |
| 3H | SJ (18.04) | **Mixed-Exp-MLM** (5.35) | SRT (20.08) | **Mixed-Exp-MLM** (5.31) |
| 6H | SRT (44.3) | **Mixed-Exp-MLM** (5.16) | SRT (41.91) | **Mixed-Exp-MLM** (4.69) |
| 12H | SRT (40.37) | **Mixed-Exp-MLM** (4.83) | SRT (43.51) | **Mixed-Exp-MLM** (4.72) |
| 1D | SJ (31.73) | **Mixed-Exp-MLM** (5.11) | SRT (32.63) | **Mixed-Exp-MLM** (4.71) |
| 5D | SJ (12.09) | **Weibull-MOM** (5.9) | SJ (13.03) | **Weibull-MOM** (5.46) |
| M | **SRT** (7.37) | Weibull-MOM (11.4) | SRT (7.7) | **Weibull-MOM** (7.14) |





**Table 7.** Performance ranking numbers for the 2-fold cross validation of regionalized precipitation amount models in the **winter season**. The underlined numbers indicate the best parametric (P) and non-parametric (NP) models. The bold numbers indicate the best overall model for each validation sample and temporal resolution.

| | 1H | 2H | 3H | 6H | 12H | 1D | 5D | M |
|---|---|---|---|---|---|---|---|---|
| **Calibration Sample 1** | | | | | | | | |
| OK - MOM | 10.45 | 8.63 | 7.87 | 7.22 | 6.27 | **_4.45_** | **_4.27_** | **_4.05_** |
| OK - MLE | 8.49 | 23.91 | 19.26 | 11.73 | 10.18 | 6.35 | 16.57 | - |
| OK - MOM DAILY | 7.83 | _4.97_ | _4.67_ | **4.21** | _4.84_ | - | - | - |
| OK - MLE DAILY | _5.54_ | 14.12 | 6.21 | 4.55 | 26.56 | - | - | - |
| PK - NP | 7.46 | 7.34 | 7.69 | 9.17 | _9.47_ | _5.31_ | 7.32 | 6.93 |
| PK - NP DAILY | **4.08** | **4.05** | _4.23_ | _5.65_ | 10.26 | - | - | - |
| NNS - MOM | 8.68 | 6.09 | _5.63_ | 5.74 | 6.16 | - | - | - |
| NN - MOM | 15.75 | 13.33 | 13.30 | 11.85 | 10.51 | 5.99 | _5.92_ | _6.34_ |
| NNS - MLE | 7.72 | _5.59_ | 5.67 | _5.59_ | _5.63_ | - | - | - |
| NN - MLE | 11.39 | 9.53 | 9.73 | 10.68 | 10.56 | 7.30 | 8.96 | 9.62 |
| NNS -NP | _6.23_ | 5.74 | 6.04 | 7.58 | 11.70 | - | - | - |
| NN - NP | 10.21 | 10.53 | 10.60 | 12.92 | 14.11 | _5.50_ | 7.26 | 287.63 |
| **Calibration Sample 2** | | | | | | | | |
| OK - MOM | 9.36 | 8.07 | 7.84 | 7.10 | 6.67 | **_4.10_** | **_4.68_** | **_4.79_** |
| OK - MLE | 6.92 | 31.26 | 28.79 | 11.16 | 8.86 | 6.11 | 6.88 | - |
| OK - MOM DAILY | 6.05 | _4.90_ | _5.10_ | **4.76** | 6.29 | - | - | - |
| OK - MLE DAILY | _5.73_ | 14.38 | 9.88 | 5.33 | **_4.59_** | - | - | - |
| PK - NP | 5.40 | 6.16 | 8.01 | 10.85 | _9.38_ | _6.75_ | 9.89 | 8.37 |
| PK - NP DAILY | **4.08** | **4.21** | _5.26_ | _7.61_ | 10.37 | - | - | - |
| NNS - MOM | 7.77 | 6.25 | 6.12 | 5.57 | 7.47 | - | - | - |
| NN - MOM | 13.95 | 13.55 | 12.53 | 11.86 | 9.58 | 5.14 | _5.95_ | _5.95_ |
| NNS - MLE | 6.61 | _5.22_ | **4.95** | _4.91_ | _5.69_ | - | - | - |
| NN - MLE | 8.88 | 8.99 | 9.16 | 10.08 | 9.13 | 7.57 | 11.21 | 9.61 |
| NNS -NP | _5.21_ | 5.36 | 6.22 | 8.11 | 11.81 | - | - | - |
| NN - NP | 8.11 | 9.65 | 11.04 | 13.53 | 12.05 | _4.72_ | 6.91 | 282.68 |




**Table 8.** Performance ranking numbers for the 2-fold cross validation of regionalized precipitation amount models in the **summer season**. The underlined numbers indicate the best parametric (P) and non-parametric (NP) models. The bold numbers indicate the best overall model for each validation sample and temporal resolution.

| | 1H | 2H | 3H | 6H | 12H | 1D | 5D | M |
|---|---|---|---|---|---|---|---|---|
| **Calibration Sample 1** | | | | | | | | |
| OK - MOM | 13.39 | 9.50 | 8.90 | 7.55 | **4.47** | **4.28** | **4.00** | **4.11** |
| OK - MLE | 9.58 | 7.12 | 6.92 | 6.52 | 65.88 | 4.44 | 12.14 | - |
| OK - MOM DAILY | 13.40 | 8.58 | 9.04 | 5.83 | 8.66 | - | - | - |
| OK - MLE DAILY | 6.53 | **4.13** | **4.06** | 7.85 | 20.38 | - | - | - |
| PK - NP | 6.83 | 8.10 | 10.32 | 11.77 | 9.35 | 10.24 | 11.65 | 6.08 |
| PK - NP DAILY | **4.09** | 5.55 | 7.36 | 9.73 | 18.03 | - | - | - |
| NNS - MOM | 13.17 | 9.67 | 9.34 | 6.95 | 9.71 | - | - | - |
| NN - MOM | 17.76 | 14.42 | 14.33 | 11.92 | 7.76 | 6.41 | 6.48 | 5.50 |
| NNS - MLE | 7.50 | 4.52 | 4.64 | **4.60** | 6.96 | - | - | - |
| NN - MLE | 14.14 | 12.45 | 11.81 | 10.82 | 7.28 | 12.04 | 12.80 | 7.79 |
| NNS -NP | 4.56 | 6.13 | 7.91 | 10.59 | 19.04 | - | - | - |
| NN - NP | 13.06 | 13.25 | 15.09 | 15.71 | 12.27 | 5.40 | 11.84 | 269.22 |
| **Calibration Sample 2** | | | | | | | | |
| OK - MOM | 11.83 | 8.00 | 8.65 | 7.14 | **4.20** | **4.12** | **4.00** | **4.00** |
| OK - MLE | 9.36 | 5.61 | 6.14 | 6.17 | 5.30 | 4.38 | 37.77 | - |
| OK - MOM DAILY | 12.29 | 8.67 | 8.39 | 6.12 | 6.69 | - | - | - |
| OK - MLE DAILY | 7.88 | **4.04** | **4.00** | **4.12** | 7.00 | - | - | - |
| PK - NP | 5.43 | 5.52 | 7.26 | 8.78 | 6.44 | 9.45 | 10.15 | 8.16 |
| PK - NP DAILY | **4.10** | 5.04 | 7.02 | 9.71 | 13.47 | - | - | - |
| NNS - MOM | 14.59 | 10.15 | 9.88 | 7.87 | 8.26 | - | - | - |
| NN - MOM | 16.83 | 11.77 | 11.59 | 9.53 | 5.94 | 5.88 | 4.83 | 4.77 |
| NNS - MLE | 8.61 | 4.99 | 5.26 | 4.61 | 6.15 | - | - | - |
| NN - MLE | 12.48 | 8.58 | 8.64 | 8.15 | 5.63 | 10.76 | 10.29 | 7.92 |
| NNS -NP | 5.45 | 6.17 | 7.76 | 10.18 | 14.00 | - | - | - |
| NN - NP | 9.97 | 9.13 | 10.63 | 10.27 | 7.75 | 5.59 | 8.57 | 260.95 |





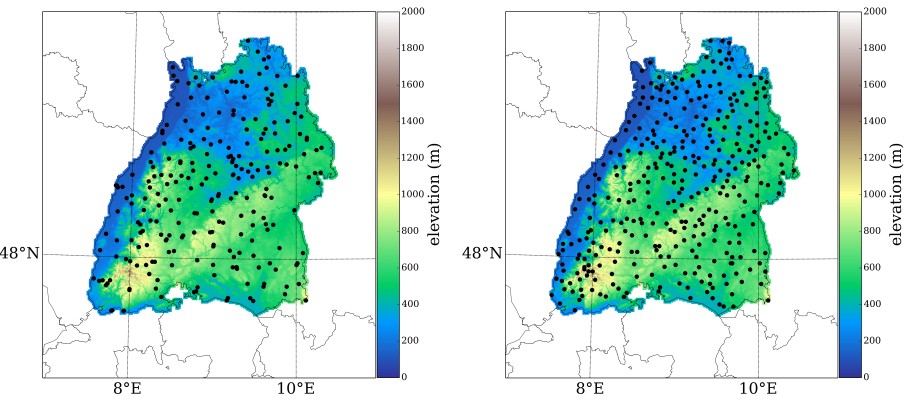

**Figure 1.** Locations of high resolution (hourly and 5 min, left) and daily rain gauges (right) in Baden-Württemberg.

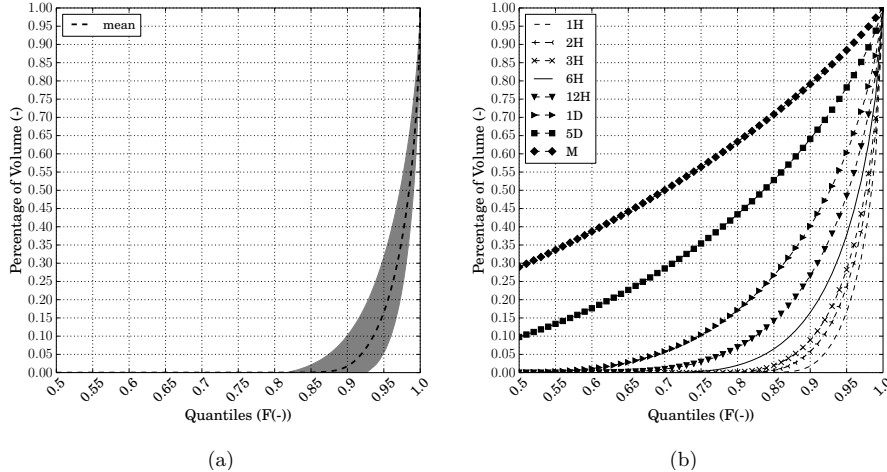

**Figure 2.** In (a) the range of the Lorenz-curves and the mean Lorenz-curve for hourly rainfall values of all rainfall gauges inside the study region are shown, in (b) the mean Lorenz-curves are shown for different temporal resolutions.




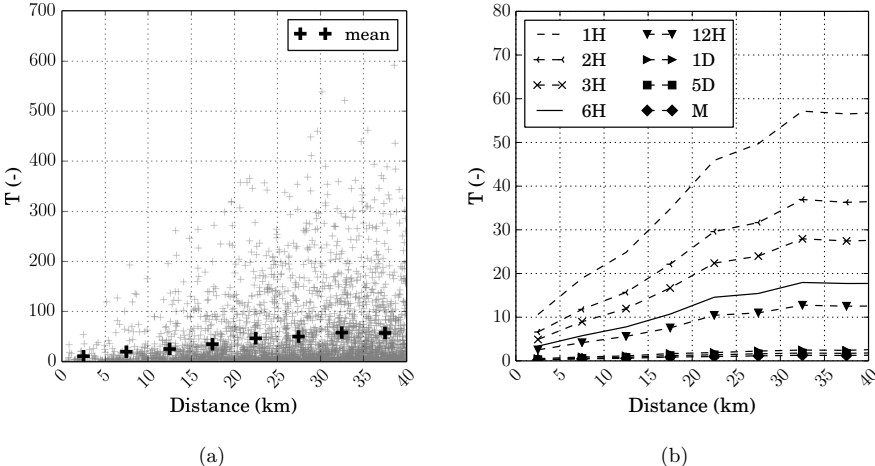

(a)                   (b)

**Figure 3.** $T$ criterion over distance: (a) shows the results for hourly distribution functions of all gauge pairs (grey crosses) and their mean calculated for 5 km classes. (b) shows the mean values of the $T$ criterion for different temporal resolutions.

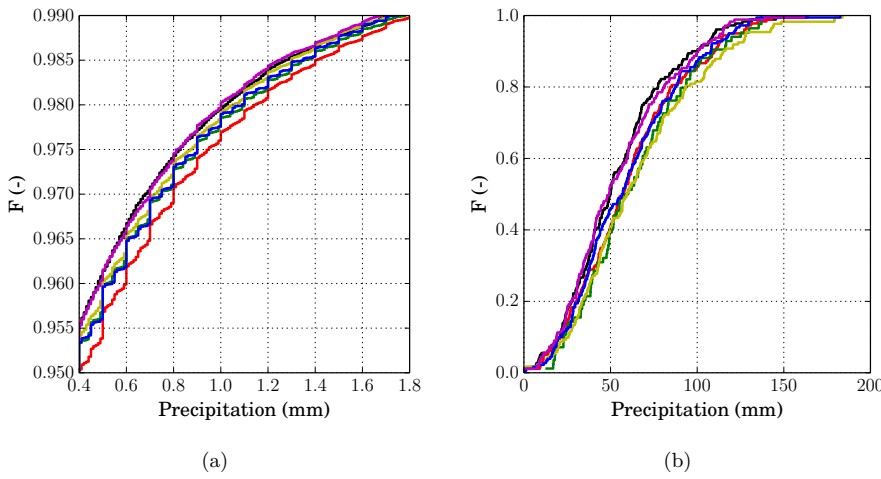

(a)                   (b)

**Figure 4.** EDFs of hourly (a) and monthly (b) precipitation amounts for gauge Stuttgart / Schnarrenberg and its five closest gauges for a quantile interval. It shows that the order of the EDFs is quite persistent over a wide quantile range for low and high resolutions. Note: As the daily and hourly data set are not the same, the colors in the two graphs do not represent the same gauges.





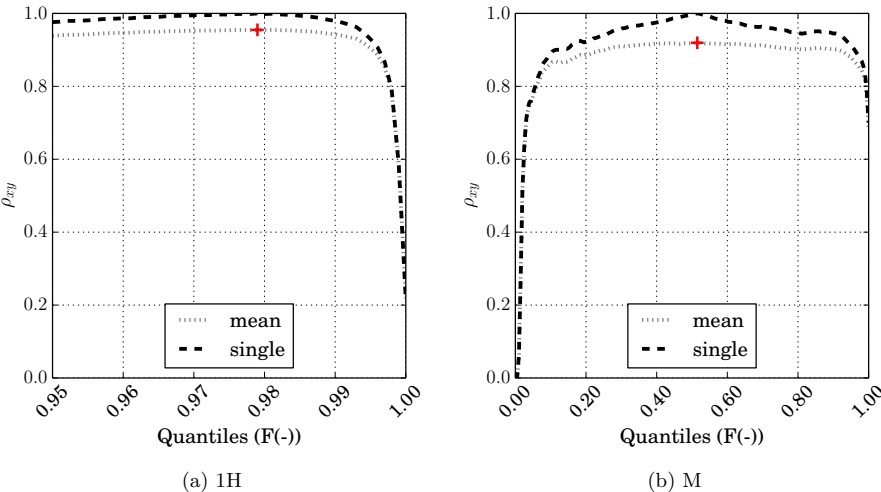

**Figure 5.** The mean rank correlations $\rho_{xy}$ of (a) hourly (1H) and (b) monthly (M) quantile values for all gauge pairs of discrete quantiles in 0.001 (1H) and 0.005 steps (M) ranging from $Q_{th}$ to 1 (gray dotted line). They are calculated to define the control quantile ($Q_c$) which exhibits the greatest mean rank correlation $\rho_{xy}$ (red cross). The black dashed line shows the (single) rank correlations $\rho_{xy}$ of quantile values at $Q_c$ (red cross) with quantile values of the remaining quantiles.

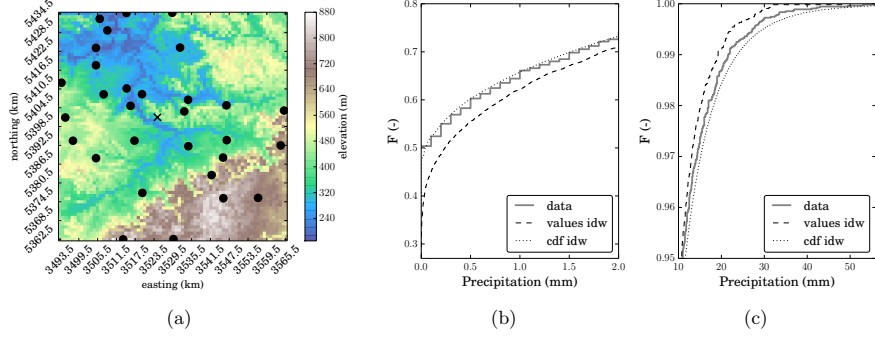

**Figure 6.** Regionalization example: (a) shows the daily target gauge (black cross) and the 30 neighboring daily gauges (black dots) of the regionalization example. In (b) and (c) parts of the EDF of the target gauge ($data$), the EDF of the interpolated rainfall values ($values_{idw}$) and the interpolated non-parametric estimate ($cdf_{idw}$) of the target CDF are depicted.





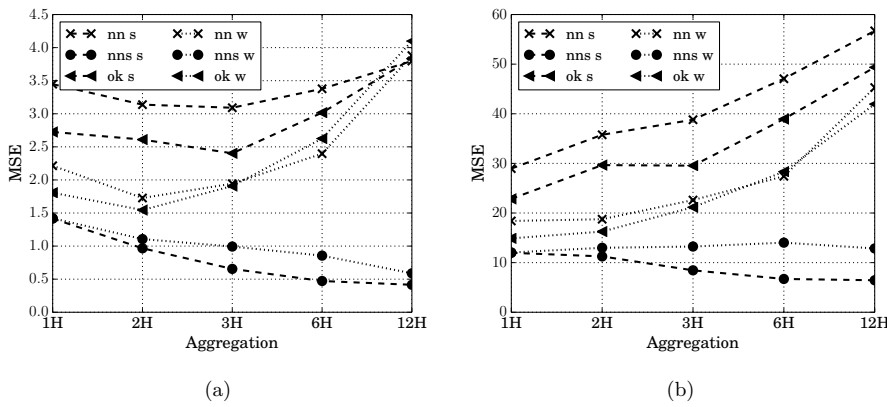

(a)                 (b)

**Figure 7.** The mean squared errors (mse) for quantile values of discrete quantiles (in 0.001 steps) greater than $Q_{th}$ (see Table 1) (a) and greater than 0.995 (b) in winter (dotted) and summer (dashed) for nearest neighbor (NN), nearest neighbor scaled (NNS) and OK of rainfall values over different aggregations. At first the mean squared error over discrete quantiles is calculated for each gauge which is followed by calculating the mean of these over the whole study region.




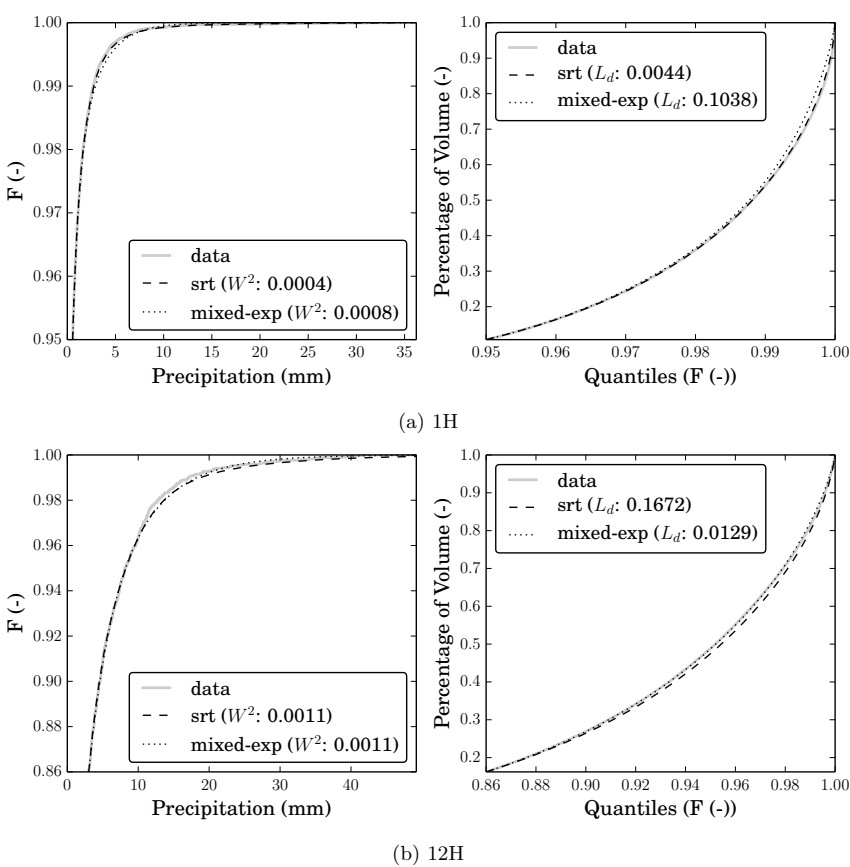

**Figure 8.** Exemplary empirical (data), non-parametric (SRT) and parametric (Mixed-Exp) CDF (left) and Lorenz-curve (right) for hourly (1H) and 12 hourly (12H) resolution of a chosen gauge. Also the values of the two quality measures $L_d$ and $W^2$ are indicated.





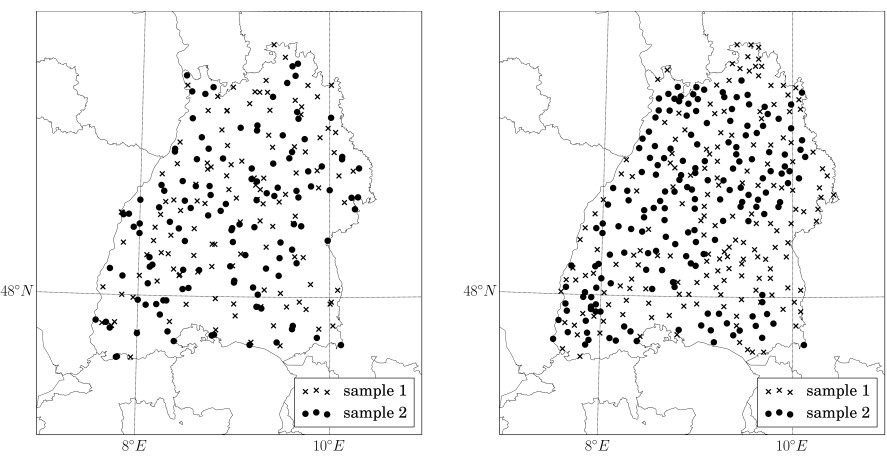

**Figure 9.** Locations of the two samples for the 2-fold cross validation of sub-daily (left) and daily gauges (right).



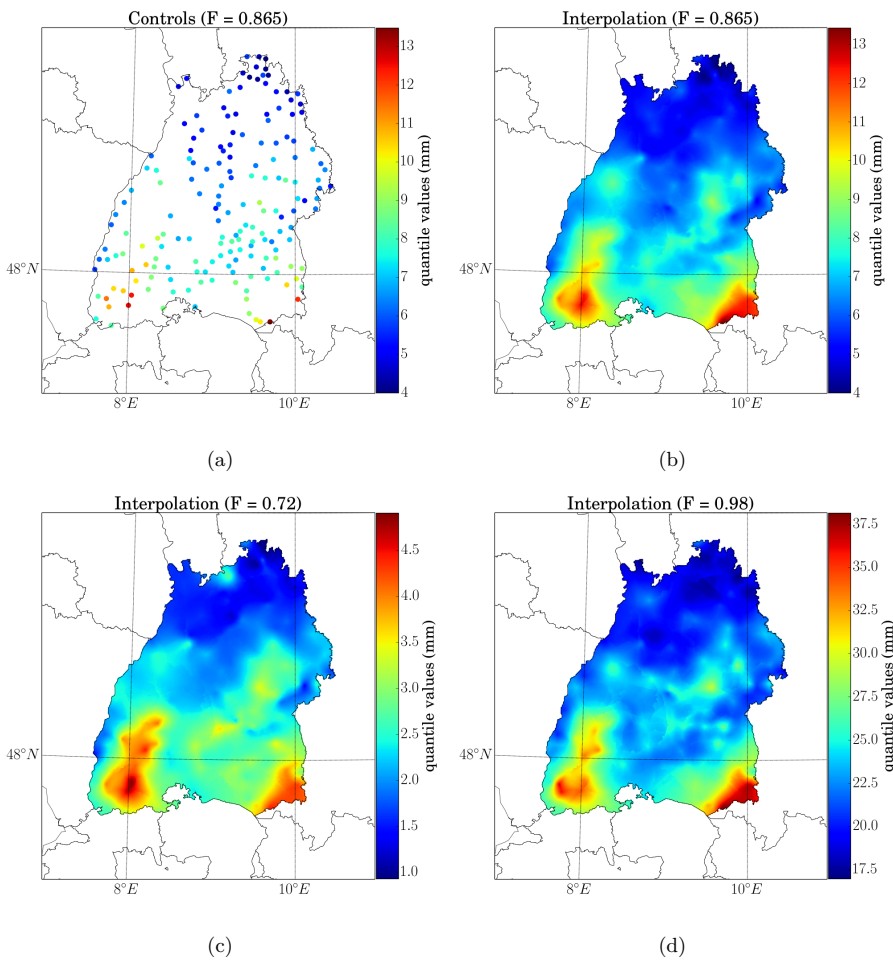

**Figure 10.** Illustrations for the kriging procedure of non-parametric distributions with daily values (1D) of the summer season using calibration sample 1 (see Fig. 9): In (a) the non-parametric $QV_c$ of the 0.865 $Q_c$ at the gauges are shown, which then lead to the interpolated values in (b) using interpolation weights $\phi_j$ resulting from PK. The same interpolation weights $\phi_j$ are used for the remaining quantiles, for which examplary results are shown in (c) for the 0.72 quantile and in (d) for the 0.98 quantile. An exponential variogram with a range of 41 $km$ and a sill of 2.2 $mm^2$ is used.