# Peer review of "Regionalizing non-parametric precipitation amount models on different temporal scales"

_Hydrology and Earth System Sciences, 2016_

## Referee Comment (RC1) · Anonymous Referee #1 · 20 Nov 2016

General comments:

The authors analyse fitting and regionalisation of parametric and non-parametric distributions for precipitation amounts considering temporal discretisations from 1 hour to 1 month. The topic is highly relevant for stochastic precipitation modelling for unobserved locations and subsequent hydrological applications like derived flood frequency analysis in mesoscale catchments.

The interpolation scheme for non-parametric distributions using control quantiles is novel. The results show that non-parametric distributions are beneficial for the short hourly time steps and that the non-parametric interpolation is working well and quite robust. Regarding parametric interpolation, the use of moments instead of parameters

seems better. In general, the utilisation of additional information from daily network is always beneficial.

The paper is well written and clear in structure. The conclusions are well supported by the analyses. However, some presentation issues mentioned below should be addressed in a revision before publication.

Detailed comments:

1. Page 2, lines 15-16: How is the interpolation of non-parametric distributions by Lall et al. (1996) done. This becomes not clear here.

2. Page 9, 10: The application of MLM and MOM for parameter estimation of distribution functions is well known. The authors may consider to remove this part with Eq. (18) to (22) and use just a reference here.

3. Page 10: Also, the estimation of the rank correlation is well known. This part with equations (23) to (25) may be removed here as well.

4. Page 12: How are the theoretical variograms fitted; which method is used; using least squares over the full range of distances? If yes, for the latter, why are close distances not given higher weights since they are more important for interpolations? Please give some more information, may be include an equation for fitting.

5. Page 14, lines 24ff: It becomes not fully clear how zero precipitation values are handled in the whole process. I thought they were excluded from the cdf's but now we have $P_0$ again? This needs to be better explained.

6. Page 15, lines 11-20: This explanation becomes not clear to me. I thought daily amounts were disaggregated according to the closest hourly proportions on the daily sum from the closest recording station? So, in this explanation for instance what means "assign the rainfall values" (line 12); what means Eq. (36); at which step are the daily data disaggregated, etc? This part needs to be much better explained.

7. Page 19: I would suggest to give some example variograms here. Also, compare and contrast the variogram parameters to the spatial persistence of T (Fig. 3).

---

## Referee Comment (RC2) · Anonymous Referee #2 · 1 Dec 2016

Paper summary

The study by Mosthaf and Bárdossy "Regionalizing non-parametric precipitation amount models on different temporal scales" establishes a comparison between different parametric models and a new proposed non-parametric method, used for regionalization of rainfall distributions over various temporal scales (ranging from 1 hour to 1 month), in the federal state of Baden-Württemberg, in Germany. Especially for sub-daily scales, the Authors consider rainfall records which are higher than predefined quantile values, to account for measurement errors associated with low rainfall intensities. The basis of the proposed approach is an observed persistence of the empirical distribution functions over different quantiles, which is apparent for all measurement

locations in the region of study. For the evaluation of each method's performance, two distribution measures are adopted, while results show that parametric approaches are overall better than the proposed non parametric one, for scales higher than 2 hours. Also, it is shown that the use of additional information from daily measurements is beneficial.

General Comment

Although the subject of the manuscript is of interest and the idea on which the proposed approach is based is new, the overall level of the manuscript and significance of work is not sufficient. Thus, a suggestion of major revisions or rejection applies. Below, I explicitly refer to each one of my concerns.

Major Comment 1

A first major concern is the excessive length of the manuscript: The text consists of 22 pages (single line spacing), while the Authors also present 8 tables and 10 figures. My suggestion is for the Authors to reduce the length of the manuscript at least 25%. Indeed, there are some sections that could be provided in the Appendix, or as supplementary information, or even completely removed. The latter is also true for some tables and figures. For example, since the article is not a review paper, sections 3, 6 8.1-8.2, and 9 should be reduced in size and some parts should be provided in the Appendix. Evidently, the proposed methodology is explained quite "late" in the text (page 13), while its application and comparison to parametric models is provided in page 20. Similarly, apart from the tables and figures associated with the above sections, table 6 essentially provides the same results as table 5. There is no need for both tables. Although, the Authors can more efficiently decide which parts of the text should be reduced in size, my point is that the present version of the manuscript is too explanatory in some cases including unnecessary information. In other words, the text should be more focused on the findings of the present work.

Major Comment 2

[Figure]

A second concern is the actual innovation and value of the presented work. Although the basis of the proposed non parametric approach is new and of potential interest, according to the obtained results, the parametric models are more effective, both in terms of point-wise estimation (Tables 4, 5 and 6) and regionalization (Tables 7 and 8). Evidently, with the only exception being the hourly rainfall, where the non parametric approach is consistently the best performing one for both samples 1 and 2, and both seasons, overall, the parametric models result in smaller distributional-related errors. Moreover, even in the case where the non parametric and the parametric methods would be of the same overall performance, the parametric approaches may again be preferred since they can be more effectively used for addressing risk and estimating rainfall extremes in periods different than the control one (i.e. 1997-2011): contrary to the non-parametric approaches, theoretical distribution models allow for more robust rainfall estimates, with approximate validity also beyond the range of the historical data in the considered control period (see Langousis et al., 2016a and references therein). That said, although the idea presented in sections 7 and 8.3 is potentially important, the results and the associated discussion in the rest sections do not support or indicate a substantial innovation or significance.

Major Comment 3

The parametric models used in this study (section 6.2), although four in number, do not include a Pareto distribution. In their conclusions, Authors mention that Pareto distributions can be also tested in the future, however, in my humble opinion, this is not sufficient. At least for the comparison section to be complete, one should include in the analysis a Pareto model (e.g. Generalized Pareto Distribution) in this study, where the proposed approach is explained and compared with other common methods. Pareto distributions have been indicated as a very efficient class for modeling daily rainfall, while towards the latter, some studies have concluded that they outperform exponential models (see Papalexiou et al., 2013; Langousis et al., 2016b and references therein).

Comment 1

In section 5, the Authors assess the consistency with which one can apply Ordinary Kriging for interpolating rainfall distributions. In doing so, they evaluate the "similarity" between different rainfall distributions for increasing distances for all temporal scales, and they present the results in Figure 3. In lines 9-10, page 6, the Authors state: "The graphs in Fig. 3 show a decreasing similarity of the distribution functions with increasing distances over all temporal resolutions, as...". However, the latter comment is not accurate for temporal scales higher or equal to 1 day. Evidently, there is no significant difference in the value of the adopted statistic even for distances on the order of 40 km. More discussion or even investigation is needed on this matter.

Comment 2

Lines 5-7, page 14: How the interpolation of the non parametric distribution functions is established to the target location? Is this done based on the new proposed approach described in section 8.3? If yes, it should be explained more explicitly. If no, then the Authors should include the approach used in section 9 (for cdf idw), in the comparison of section 11.3.2.

Comment 3

Lines 4-6, page 18: The readers are not able to validate these statements. The rankings provided in tables 5 and 6, are combined, i.e. they summarize the performance of each model based on both criteria (38) and (40). Due the latter, the discussion of Figure 8 can be challenged as well.

Comment 4

Table 5 shows remarkably high errors (about 5000) in the performance of the exponential and mixed exponential models in case of the monthly precipitation amounts, for both seasons. Considering that Table 5 corresponds to wise point estimation, such high errors indicate complete inconsistency in the fitting of each model to the data, which is not reasonable to me, especially if on considers that in the case of 5-days

scale, the corresponding errors are on the order of 30. The Authors should discuss this result.

Comment 5

Line 15, page 9: Instead of MOM, why not using L-moments estimator (PWM)?

Comment 6

Concerning the level of writing, apart from the length of the manuscript which has already been discussed (see major comment 1), there numerous cases of ambiguity and typos, which means that the Authors need to refine their text. Below, I mention just a few examples: 1) line 25, page 1: "In order to run. . .for these sites.". Ambiguous sentence, please rephrase. 2) line 25, page 3: "only gauges with. . . are chosen.". Please explain better. 3) equation (1): since this equation refers to precipitation amount (see line 13 in the same page), please replace $-\infty \leq x < \infty$ with $0 \leq x < \infty$. 4) line 9, page 5 (and in other points throughout the manuscript): please replace "0.95 Qth" with "Qth =0.95". 5) lines 16-17, page 5: "85% is defined. . ." Ambiguous sentence, please rephrase. 6) line 9, page 5: "between 0.2 and 1.7 mm". This is inconsistent with the corresponding value in Table 1.

References

Langousis, A., A. Mamalakis, M. Puliga and R. Deidda (2016b) Threshold detection for the generalized Pareto distribution: Review of representative methods and application to the NOAA NCDC daily rainfall database, Water Resour. Res., 52, doi:10.1002/2015WR018502.

Langousis, A., A. Mamalakis, R. Deidda and M. Marrocu (2016a) Assessing the relative effectiveness of statistical downscaling and distribution mapping in reproducing rainfall statistics based on climate model results, Water Resour. Res., 52, doi:10.1002/2015WR017556.

Papalexiou, S.M., D. Koutsoyiannis and C. Makropoulos (2013) How extreme is

extreme? An assessment of daily rainfall distribution tails, Hydrol. Earth Syst. Sci., 17(2), 851-862.

Please also note the supplement to this comment:
http://www.hydrol-earth-syst-sci-discuss.net/hess-2016-458/hess-2016-458-RC2-supplement.pdf

———————————————————

---

## Referee Comment (RC3) · Anonymous Referee #3 · 4 Dec 2016

In this paper the authors used parametric and non-parametric rainfall models for modeling precipitation amounts at different temporal resolutions varying from hourly to monthly. In the non-parametric case the authors proposed a new interpolation scheme, by comparing the Empirical Distribution Functions for all locations and defining a control quantile, which exhibits the greatest mean rank correlation among all pairwise rank correlation of quantile values of all gauge pairs. Extensive comparison with parametric models is also presented in this research, and the aim is to obtain precipitation amount distributions at ungagged locations.

Comment 1

It is not clear why the authors use the Inverse Distance Weighting method in section 9 if

they first talk about Ordinary Kriging as a regionalization method in section 8. Content from section 9 is a confusing. At the end of this section the authors said the following: (the method cdfidw) it will be adopted in the sequel with OK as interpolation technique. It is not clear what this statement means.

Comment 2

Some of the details in section 6.2 (parametric models) can be skipped since they are very well known results. The same situation can be said from section 8.1. The paper is rather long and the shortening of this section would help with the final length of the paper.

Minor comments

Page 5, par 25: a comma should have inserted after to be preassigned

Page 7, par 25. Instead of is investigated it should say are investigated

Figure 2: The y axis labels are not percentages as the title suggests

Figure 5: It is not clear what does the legend "single" mean.
* * *

---

## Author Comment (AC1) · 8 Dec 2016

We would like to thank the three anonymous reviewers for their informative comments. Two common concerns of the reviewers are the length of the manuscript and the missing clarity in section 9. We will work on these two concerns and believe that this will certainly improve the quality of the manuscript. Our answers to the individual reviewers comments follow below:

**Reviewer 1**

[Figure]

1. Page 2, lines 15-16: How is the interpolation of non-parametric distributions by Lall et al. (1996) done. This becomes not clear here.

Answer: Lall et al. (1996) did not interpolate non-parametric distributions, they just mentioned that no interpolation scheme does exist. Such a scheme needs to be different than parametric interpolation schemes, as non-parametric distributions do not have any parameter.

2. Page 9, 10: The application of MLM and MOM for parameter estimation of distribution functions is well known. The authors may consider to remove this part with Eq. (18) to (22) and use just a reference here.

Answer: This part will be shortened or removed.

3. Page 10: Also, the estimation of the rank correlation is well known. This part with equations (23) to (25) may be removed here as well.

Answer: This part will be shortened or removed.

4. How are the theoretical variograms fitted; which method is used; using least squares over the full range of distances? If yes, for the latter, why are close distances not given higher weights since they are more important for interpolations? Please give some more information, may be include an equation for fitting.

Answer: The theoretical variograms are fitted using least squares with distances up to 100 km, grouping the empirical variogram values into distance classes of 10 km. In the attachment figuresvariograms.pdf the variograms of different parameters for temporal

resolutions of 1H and 12H are shown for the winter and summer season of calibration sample 2. The black crosses represent the empirical variogram values of the 10 km distance classes, which are used for the least squares fit. The grey crosses represent the empirical variogram values of 1 km distance classes. When a good fit is obtained with the 10 km distance classes, the theoretical variograms also seem to represent the smaller distances quite well, therefore, we think higher weights for close distances are not necessary. Sometimes the empirical variogram of the considered parameter exhibits a close to pure nugget effect (e.g. $\lambda_2$ of P-Mixed-EXP-MLM Calibration Sample 2, winter season, 12H), then the fit of the variogram was not good, however, this not necessarily leads to bad regionalization results (see Table 7 in the manuscript).

5. Page 14, lines 24ff: It becomes not fully clear how zero precipitation values are handled in the whole process. I thought they were excluded from the cdf's but now we have P0 again? This needs to be better explained.

Answer: Only in section 9 zero values were included to show the advantages of interpolating distributions instead of precipitation values regarding the value of P0, to motivate the whole process of interpolating distributions, what was not done in any previous paper to our knowledge. Zero values can be included within the interpolation of non-parametric distributions by applying the following steps. (i) Fitting a distribution to all precipitation values at each gauge. (ii) Estimate the quantile values for certain quantiles (non-exceedance probabilities) over the whole probability range (0-1) with the inverse of the fitted distributions at each gauge. (iii) Use the interpolation weights from inverse distance weighting to interpolate the quantile values of different gauges for each chosen quantile. (iv) If the quantile is below p0 for some (or all) gauges, the quantile value at these gauges will be 0 mm, which then are just included in the interpolation. (v) The highest quantile with 0 mm at the target gauge defines p0 at the target.

6. Page 15, lines 11-20: This explanation becomes not clear to me. I thought daily amounts were disaggregated according to the closest hourly proportions on the daily sum from the closest recording station? So, in this explanation for instance what means "assign the rainfall values" (line 12); what means Eq. (36); at which step are the daily data disaggregated, etc? This part needs to be much better explained.

Answer: We think you understood it correctly, the procedure works as you explained. "assign the rainfall values" just means allocating hourly values from the closest high resolution gauge to the daily target gauge and Eq. (36) is the rescaling factor calculated from the daily values of the target gauge and the daily sum of the hourly values of the closest high resolution gauge. This rescaling factor changes from day to day and simply assures that daily sums of the disaggregated hourly values at the target gauge equal the daily values measured at the target.

7. Page 19: I would suggest to give some example variograms here. Also, compare and contrast the variogram parameters to the spatial persistence of T (Fig. 3).

Answer: It is difficult to compare the spatial persistence of T with the spatial persistence of the different distribution parameters, as T considers the whole distribution function and the distribution parameters only describe the distributions in parts. However, the range of T was about 35 km, which can also be observed for some of the parameters, especially the mean of P-Weibull-MOM, $QV_c$ of NP-SRT and $QV_th$ (see attachment figuresvariograms.pdf).

**Reviewer 2**

Major Comment 1: A first major concern is the excessive length of the manuscript: The text consists of 22 pages (single line spacing), while the Authors also present 8

tables and 10 figures. My suggestion is for the Authors to reduce the length of the manuscript at least 25 %. Indeed, there are some sections that could be provided in the Appendix, or as supplementary information, or even completely removed. The latter is also true for some tables and figures. For example, since the article is not a review paper, sections 3, 6 8.1-8.2, and 9 should be reduced in size and some parts should be provided in the Appendix. Evidently, the proposed methodology is explained quite "late" in the text (page 13), while its application and comparison to parametric models is provided in page 20. Similarly, apart from the tables and figures associated with the above sections, table 6 essentially provides the same results as table 5. There is no need for both tables. Although, the Authors can more efficiently decide which parts of the text should be reduced in size, my point is that the present version of the manuscript is too explanatory in some cases including unnecessary information. In other words, the text should be more focused on the findings of the present work.

Answer: We agree with the reviewer that the manuscript is quite long and that some parts could be removed to better focus on the actual findings. We think reducing the size of sections 3, 6 and 8.1-8.2 is a good proposition, as they only provide information about already existing models and well known theory. Removing table 6 is also a good suggestion. However, we think the order of the sections should be kept, as the explanations in the successive sections are based on each other.

Major Comment 2 : A second concern is the actual innovation and value of the presented work. Although the basis of the proposed non parametric approach is new and of potential interest, according to the obtained results, the parametric models are more effective, both in terms of point-wise estimation (Tables 4, 5 and 6) and regionalization (Tables 7 and 8). Evidently, with the only exception being the hourly rainfall, where the non parametric approach is consistently the best performing one for both samples 1 and 2, and both seasons, overall, the parametric models result in smaller distributional-related errors. Moreover, even in the case where the non

parametric and the parametric methods would be of the same overall performance, the parametric approaches may again be preferred since they can be more effectively used for addressing risk and estimating rainfall extremes in periods different than the control one (i.e. 1997-2011): contrary to the non-parametric approaches, theoretical distribution models allow for more robust rainfall estimates, with approximate validity also beyond the range of the historical data in the considered control period (see Langousis et al., 2016a and references therein). That said, although the idea presented in sections 7 and 8.3 is potentially important, the results and the associated discussion in the rest sections do not support or indicate a substantial innovation or significance.

Answer: We disagree with the first part of this comment, where the reviewer proposes not to publish this manuscript, because the non parametric method only performed best for the hourly resolution. If we only had shown results for the hourly distribution, this statement would possibly have been vice versa. However, we presented the results for several temporal resolutions, as we also wanted to present the deficiencies of the newly developed non-parametric method. Even if the method performed worse over all temporal resolutions, we would consider it as important to publish the method. This may prevent the investigation of this method by another hydrologist and further more the methodology could be applied to distributions corresponding to other variables (where e.g. multi modal distributions are present). Additionally, we have shown that daily gauges are of great use for the interpolation of sub-daily distributions. The philosophy of only allowing methods for publication, which always perform best, may lead to *cherry picking* of the results and prevent an open discussion in science.

Regarding the estimation of rainfall extremes, non-parametric kernel density estimations may exhibit problems. However, using a Gaussian kernel also allows for extrapolation beyond the range of the historical data, which still needs to be evaluated. The study mentioned from the reviewer (Langousis et al., 2016a) investigates daily rainfall extremes, but not, how it is for different temporal resolutions? Also more investigations

are required to answer this question. In addition, depending on the application, rainfall extremes do not always have such a decisive character. An example is real-time control of sewer systems, where average and larger values are more important, as rainfall extremes can not be controlled by the system anyway.

Major Comment 3 : The parametric models used in this study (section 6.2), although four in number, do not include a Pareto distribution. In their conclusions, Authors mention that Pareto distributions can be also tested in the future, however, in my humble opinion, this is not sufficient. At least for the comparison section to be complete, one should include in the analysis a Pareto model (e.g. Generalized Pareto Distribution) in this study, where the proposed approach is explained and compared with other common methods. Pareto distributions have been indicated as a very efficient class for modeling daily rainfall, while towards the latter, some studies have concluded that they outperform exponential models (see Papalexiou et al., 2013; Langousis et al., 2016b and references therein).

Answer: In the references mentioned by the reviewer, the focus lies on extremes of daily rainfall, whereas in our investigations we only exclude very small rainfall values for each aggregation due to measurement errors and minor importance (see Table 1 in the manuscript). Additionally the focus of the manuscript lies on regionalization, which can influence the performance of a theoretical distribution and was to our knowledge not yet investigated for the whole range of daily rainfall values using Pareto type distributions. However, Pareto type distributions are very interesting and their regionalization performance could be looked at in a different paper.

Comment 1 : In section 5, the Authors assess the consistency with which one can apply Ordinary Kriging for interpolating rainfall distributions. In doing so, they evaluate the "similarity" between different rainfall distributions for increasing distances for all

temporal scales, and they present the results in Figure 3. In lines 9-10, page 6, the Authors state: "The graphs in Fig. 3 show a decreasing similarity of the distribution functions with increasing distances over all temporal resolutions, as. . .". However, the latter comment is not accurate for temporal scales higher or equal to 1 day. Evidently, there is no significant difference in the value of the adopted statistic even for distances on the order of 40 km. More discussion or even investigation is needed on this matter.

Answer: The reviewer is right, with the graphs in Fig. 3 a decreasing similarity of the distribution functions with increasing distances can not be verified for temporal scales higher or equal to 1 day. This is, however, just an effect of the chosen plotting scale of T. In the attachment Tdistance.pdf a graph similar to Fig. 3 b shows the development of T over distances for higher aggregations with a different plotting scale of T. With this graph our statement in the manuscript can be verified.

Comment 2 : Lines 5-7, page 14: How the interpolation of the non parametric distribution functions is established to the target location? Is this done based on the new proposed approach described in section 8.3? If yes, it should be explained more explicitly. If no, then the Authors should include the approach used in section 9 (for cdf idw), in the comparison of section 11.3.2.

Answer: In section 9 *Regionalization example* the only difference compared to the proposed approach in section 8.3 is the calculation of the weights. In section 9, they are estimated with IDW. There are several reasons why IDW instead of OK is used in section 9 to calculate the weights. Using OK with daily precipitation values leads to the additional challenge of including zero rainfall values within the estimation of the variogram and the kriging itself. As the focus of the manuscript does not lie on interpolating rainfall values, the simpler IDW method is used. Therefore, IDW is also used for the interpolation of the distributions ($cdf_{idw}$) to assure that the better performance of $cdf_{idw}$

does not originate from the calculation of the weights (different with OK and IDW), but from the chosen interpolation scheme ($cdf_{idw}$, $values_{idw}$). In the subsequent sections, IDW is not used anymore, because OK is considered as a better interpolation method than the simpler IDW. We realized that the use of IDW in this section is quite confusing and the reasons should therefore be explained in more detail.

Comment 3 : Lines 4-6, page 18: The readers are not able to validate these statements. The rankings provided in tables 5 and 6, are combined, i.e. they summarize the performance of each model based on both criteria (38) and (40). Due the latter, the discussion of Figure 8 can be challenged as well.

Answer: We agree with the reviewer that the mentioned statement can not be verified by table 5 nor table 6. However, in the attachment tablesranking.pdf we provide the required tables, with ranking numbers constructed of the mean and median separately for each quality measure. These tables verify the criticized statement, as the non-parametric methods always perform better concerning the $W^2$ measure and the parametric mostly perform better concerning the $L_d$ measure. Due to the length of the manuscript we didn't provide these tables, but both tables could be provided in an appendix of the final paper.

Comment 4 : Table 5 shows remarkably high errors (about 5000) in the performance of the exponential and mixed exponential models in case of the monthly precipitation amounts, for both seasons. Considering that Table 5 corresponds to wise point estimation, such high errors indicate complete inconsistency in the fitting of each model to the data, which is not reasonable to me, especially if on considers that in the case of 5-days scale, the corresponding errors are on the order of 30. The Authors should discuss this result.

Answer: The exponential distributions have more problems with symmetric distributions than the rest of the considered distribution types. Monthly rainfall values are quite symmetric, whereas the regarded range of the 5-days values are still right skewed. Looking at the graphs in attachment cdfs.pdf may illustrate these problems. Therefore, we do not consider the fitting of the models to the data as inconsistent.

Comment 5 : Line 15, page 9: Instead of MOM, why not using L-moments estimator (PWM)?

Answer: To our knowledge L-moments have not yet been tested for the regionalization of precipitation distributions respecting the whole range of rainfall values, that is why we preferred moments as they were already tested for regionalization.

Comment 6 : Concerning the level of writing, apart from the length of the manuscript which has already been discussed (see major comment 1), there numerous cases of ambiguity and typos, which means that the Authors need to refine their text. Below, I mention just a few examples: 1) line 25, page 1: "In order to run. . .for these sites.". Ambiguous sentence, please rephrase. 2) line 25, page 3: "only gauges with. . . are chosen.". Please explain better. 3) equation (1): since this equation refers to precipitation amount (see line 13 in the same page), please replace $-\infty \leq x < \infty$ with $0 \leq x < \infty$. 4) line 9, page 5 (and in other points throughout the manuscript): please replace "0.95 Qth" with "Qth =0.95". 5) lines 16-17, page 5: "85% is defined. . ." Ambiguous sentence, please rephrase. 6) line 9, page 5: "between "between 0.2 and 1.7 mm". This is inconsistent with the corresponding value in Table 1.

Answer: The text of the manuscript indeed needs further refinement.

**Reviewer 3**

[Figure]

Comment 1 : It is not clear why the authors use the Inverse Distance Weighting method in section 9 if they first talk about Ordinary Kriging as a regionalization method in section 8. Content from section 9 is a confusing. At the end of this section the authors said the following: (the method cdfidw) it will be adopted in the sequel with OK as interpolation technique. It is not clear what this statement means.

Answer: See answer on comment 2 of the second reviewer.

Comment 2 : Some of the details in section 6.2 (parametric models) can be skipped since they are very well known results. The same situation can be said from section 8.1. The paper is rather long and the shortening of this section would help with the final length of the paper.

Answer: We agree with the reviewer and will remove some of the details in the mentioned sections.

Minor comments:
Page 5, par 25: a comma should have inserted after to be preassigned
Page 7, par 25. Instead of is investigated it should say are investigated
Figure 2: The y axis labels are not percentages as the title suggests
Figure 5: It is not clear what does the legend "single" mean

Answer: These minor corrections will be made.

Please also note the supplement to this comment:
http://www.hydrol-earth-syst-sci-discuss.net/hess-2016-458/hess-2016-458-AC1-supplement.zip

---

## Referee Report (RR1)

Review #2 of

"Regionalizing non-parametric precipitation amount models on different temporal scales"

submitted to *Hydrology and Earth System Sciences*

March 2017

My initial suggestion for the manuscript "Regionalizing non-parametric precipitation amount models on different temporal scales" was "major revisions" accompanied with nine comments. The authors addressed some of my comments adequately. Major comments 2 and 3 were not addressed. Below, I comment on authors' responses (red-colored text) concerning the two unaddressed comments.

**Comment 1**

**Reviewer 2:**

*Major Comment 2: A second concern is the actual innovation and value of the presented work. Although the basis of the proposed non parametric approach is new and of potential interest, according to the obtained results, the parametric models are more effective, both in terms of point-wise estimation (Tables 4, 5 and 6) and regionalization (Tables 7 and 8). Evidently, with the only exception being the hourly rainfall, where the non parametric approach is consistently the best performing one for both samples 1 and 2, and both seasons, overall, the parametric models result in smaller distributional-related errors. Moreover, even in the case where the non parametric and the parametric methods would be of the same overall performance, the parametric approaches may again be preferred since they can be more effectively used for addressing risk and estimating rainfall extremes in periods different than the control one (i.e. 1997-2011): contrary to the non-parametric approaches, theoretical distribution models allow for more robust rainfall estimates, with approximate validity also beyond the range of the historical data in the considered control period (see Langousis et al., 2016a and references therein). That said, although the idea presented in sections 7 and 8.3 is potentially important, the results and the associated discussion in the rest sections do not support or indicate a substantial innovation or significance.*

**Authors:**

*Answer: We disagree with the first part of this comment, where the reviewer proposes not to publish this manuscript, because the non parametric method only performed best for the hourly resolution. If we only had shown results for the hourly distribution, this statement would possibly have been vice versa. However, we presented the results for several temporal resolutions, as we also wanted to present the deficiencies of the newly developed non-parametric method. Even if the method performed worse over all temporal resolutions, we would consider it as important to publish the method. This may prevent the investigation of this method by another hydrologist and further more the methodology could be applied to distributions corresponding to other variables (where e.g. multi modal distributions are present). Additionally, we have shown that daily gauges are of great use for the interpolation of sub-daily distributions. The philosophy of only allowing methods for publication, which always perform best, may lead to cherry picking of the results and prevent an open discussion in science. Regarding the estimation of rainfall extremes, non-parametric kernel density estimations may exhibit problems. However, using a Gaussian kernel also allows for extrapolation beyond the range of the historical data, which still needs to be evaluated. The study mentioned from the reviewer (Langousis et al., 2016a) investigates daily rainfall extremes, but not, how it is for different temporal resolutions? Also more investigations are required to answer this question. In addition, depending on the application, rainfall extremes*

*do not always have such a decisive character. An example is real-time control of sewer systems, where average and larger values are more important, as rainfall extremes cannot be controlled by the system anyway.*

**Reviewer 2:**

I agree with some of the arguments stated in this paragraph. However, my official suggestion was not "rejection of the manuscript". I suggested "major revisions". The reason for this is that although I had major concerns about the level of the manuscript, in terms of innovation and presentation efficiency (see my nine comments in the first round of revisions), I recognized and also indicated the potential importance of authors' results. With that being said, authors' statement "*We disagree with the first part of this comment, where the reviewer proposes not to publish this manuscript…*", and more importantly their whole discussion on my philosophy (towards what is worth publishing and what is not) are not based on my actual and official suggestion. The authors should not so easily jump into conclusions and judge a reviewer's judgment or philosophy based on their assumptions about reviewer's opinion, and not based on his/her actual and official suggestion.

The authors also state: *Even if the method performed worse over all temporal resolutions, we would consider it as important to publish the method.*
This is only authors' opinion. A reviewer needs to point out all possible shortcomings of a proposed method. The final decision will be made by the handling Editor.

Concerning the technical part, in their response the authors state: *The study mentioned from the reviewer (Langousis et al., 2016a) investigates daily rainfall extremes, but not, how it is for different temporal resolutions?*
The study I mentioned refers to daily rainfall but not only to rainfall extremes. Also, note that both references provided by the authors themselves (see their conclusions) refer to daily rainfall (not to finer temporal scales), and consider the use of mixed Pareto-type distributions (see also my next comment).

**Comment 2**

**Reviewer 2:**

*Major Comment 3: The parametric models used in this study (section 6.2), although four in number, do not include a Pareto distribution. In their conclusions, Authors mention that Pareto distributions can be also tested in the future, however, in my humble opinion, this is not sufficient. At least for the comparison section to be complete, one should include in the analysis a Pareto model (e.g. Generalized Pareto Distribution) in this study, where the proposed approach is explained and compared with other common methods. Pareto distributions have been indicated as a very efficient class for modeling daily rainfall, while towards the latter, some studies have concluded that they outperform exponential models (see Papalexiou et al., 2013; Langousis et al., 2016b and references therein).*

**Authors:**

*Answer: In the references mentioned by the reviewer, the focus lies on extremes of daily rainfall, whereas in our investigations we only exclude very small rainfall values for each aggregation due to measurement errors and minor importance (see Table 1 in the manuscript). Additionally the focus of the manuscript lies on regionalization, which can influence the performance of a theoretical distribution and was to our knowledge not yet investigated for the whole range of daily rainfall values using Pareto type distributions. However, Pareto type distributions are very interesting and their regionalization performance could be looked at in a different paper.*

**Reviewer 2:**

In a recent study (see doi: 10.1002/2016WR019578) a parametric approach for simultaneous bias correction and regionalization of climate model rainfall is proposed based on the use of GPD above a certain threshold (mixed type). It is proved that it outperforms the nonparametric alternative.
In any case though, since the authors themselves think that Pareto type distributions are very interesting and their regionalization performance should be looked, I do not see the reason that they are unwilling to add a Pareto type distribution in their analysis. Their current investigation may be regarded incomplete.

**General comment:**

In their responses, the authors either commented on my philosophy (based on what they think I think of their work, and not on my actual suggestion), or they only stated that Pareto-type distributions are potentially interesting. Yet, they are unwilling to include a Pareto-type distribution in their analysis. I consider both my comments unaddressed. I acknowledge that most of my other comments are addressed, thus, I change my suggestion from "major revisions" to "moderate revisions". A second round of revisions is needed.

---

## Author Response (AR2)

**1 Response to Review # 2**

We would like to thank the second reviewer for his comments. One of his major concerns was the exclusion of a Pareto-type distribution. In the updated manuscript a Pareto-type distribution is included to complete the manuscript.

Our answers to the individual reviewer comments follow below (blue-colored text). :

**Comment 1**

**Reviewer 2:**

*Major Comment 2: A second concern is the actual innovation and value of the presented work. Although the basis of the*
10 *proposed non parametric approach is new and of potential interest, according to the obtained results, the parametric models are more effective, both in terms of point-wise estimation (Tables 4, 5 and 6) and regionalization (Tables 7 and 8). Evidently, with the only exception being the hourly rainfall, where the non parametric approach is consistently the best performing one for both samples 1 and 2, and both seasons, overall, the parametric models result in smaller distributional-related errors. Moreover, even in the case where the non parametric and the parametric methods would be of the same overall performance,*
15 *the parametric approaches may again be preferred since they can be more effectively used for addressing risk and estimating rainfall extremes in periods different than the control one (i.e. 1997-2011): contrary to the non-parametric approaches, theoretical distribution models allow for more robust rainfall estimates, with approximate validity also beyond the range of the historical data in the considered control period (see Langousis et al., 2016a and references therein). That said, although the idea presented in sections 7 and 8.3 is potentially important, the results and the associated discussion in the rest sections do*
20 *not support or indicate a substantial innovation or significance.*

**Authors:**

1. Answer: *We disagree with the first part of this comment, where the reviewer proposes not to publish this manuscript, because the non parametric method only performed best for the hourly resolution. If we only had shown results for the hourly distribu-*
25 *tion, this statement would possibly have been vice versa. However, we presented the results for several temporal resolutions, as we also wanted to present the deficiencies of the newly developed non-parametric method. Even if the method performed worse over all temporal resolutions, we would consider it as important to publish the method. This may prevent the investigation of this method by another hydrologist and further more the methodology could be applied to distributions corresponding to other variables (where e.g. multi modal distributions are present). Additionally, we have shown that daily gauges are of great use for*
30 *the interpolation of sub-daily distributions. The philosophy of only allowing methods for publication, which always perform best, may lead to cherry picking of the results and prevent an open discussion in science. Regarding the estimation of rainfall extremes, non-parametric kernel density estimations may exhibit problems. However, using a Gaussian kernel also allows for extrapolation beyond the range of the historical data, which still needs to be evaluated. The study mentioned from the reviewer (Langousis et al., 2016a) investigates daily rainfall extremes, but not, how it is for different temporal resolutions? Also more*
35 *investigations are required to answer this question. In addition, depending on the application, rainfall extremes do not always have such a decisive character. An example is real-time control of sewer systems, where average and larger values are more important, as rainfall extremes cannot be controlled by the system anyway.*

**Reviewer 2:**
40 I agree with some of the arguments stated in this paragraph. However, my official suggestion was not "rejection of the manuscript". I suggested "major revisions". The reason for this is that although I had major concerns about the level of the manuscript, in terms of innovation and presentation efficiency (see my nine comments in the first round of revisions), I recognized and also indicated the potential importance of authors' results. With that being said, authors' statement "*We disagree with the first part of this comment, where the reviewer proposes not to publish this manuscript. . .*", and more importantly their
45 whole discussion on my philosophy (towards what is worth publishing and what is not) are not based on my actual and official suggestion. The authors should not so easily jump into conclusions and judge a reviewer's judgment or philosophy based on

their assumptions about reviewer's opinion, and not based on his/her actual and official suggestion.

The authors also state: *Even if the method performed worse over all temporal resolutions, we would consider it as important to publish the method.* This is only authors' opinion. A reviewer needs to point out all possible shortcomings of a proposed method. The final decision will be made by the handling Editor.

Concerning the technical part, in their response the authors state: *The study mentioned from the reviewer (Langousis et al., 2016a) investigates daily rainfall extremes, but not, how it is for different temporal resolutions?* The study I mentioned refers to daily rainfall but not only to rainfall extremes. Also, note that both references provided by the authors themselves (see their conclusions) refer to daily rainfall (not to finer temporal scales), and consider the use of mixed Pareto-type distributions (see also my next comment).

**Authors:**
2. Answer: *A Pareto-type distribution is included in the updated manuscript. Additional comments on the usage of non-parametric models are added to the conclusion.*

**Comment 2**

**Reviewer 2:**
*Major Comment 3: The parametric models used in this study (section 6.2), although four in number, do not include a Pareto distribution. In their conclusions, Authors mention that Pareto distributions can be also tested in the future, however, in my humble opinion, this is not sufficient. At least for the comparison section to be complete, one should include in the analysis a Pareto model (e.g. Generalized Pareto Distribution) in this study, where the proposed approach is explained and compared with other common methods. Pareto distributions have been indicated as a very efficient class for modeling daily rainfall, while towards the latter, some studies have concluded that they outperform exponential models (see Papalexiou et al., 2013; Langousis et al., 2016b and references therein).*

**Authors:**
1. Answer: *In the references mentioned by the reviewer, the focus lies on extremes of daily rainfall, whereas in our investigations we only exclude very small rainfall values for each aggregation due to measurement errors and minor importance (see Table 1 in the manuscript). Additionally the focus of the manuscript lies on regionalization, which can influence the performance of a theoretical distribution and was to our knowledge not yet investigated for the whole range of daily rainfall values using Pareto type distributions. However, Pareto type distributions are very interesting and their regionalization performance could be looked at in a different paper.*

**Reviewer 2:**
In a recent study (see doi: 10.1002/2016WR019578) a parametric approach for simultaneous bias correction and regionalization of climate model rainfall is proposed based on the use of GPD above a certain threshold (mixed type). It is proved that it outperforms the nonparametric alternative. In any case though, since the authors themselves think that Pareto type distributions are very interesting and their regionalization performance should be looked, I do not see the reason that they are unwilling to add a Pareto type distribution in their analysis. Their current investigation may be regarded incomplete.

**Authors:**
2. Answer: *A Pareto-type distribution is included in the updated manuscript.*

**2 Additional changes in the manuscript**

*The implementation of calculating the two performance measures was found to be incorrect. Therefore, the whole point wise estimation and regionalization of the precipitation amount models was repeated. This led to different results with slightly different conclusions (see corrections throughout the manuscript). The tables 3-6 and S2-S3 as well as the figures 6 and S4-S7 are changed accordingly.*

[revised manuscript text omitted]